# ARF1 prevents aberrant type I interferon induction by regulating STING activation and recycling

Maximilian Hirschenberger[1], Alice Lepelley[2], Ulrich Rupp[3], Susanne Klute[1], Victoria Hunszinger[1], Lennart Koepke[1], Veronika Merold[4], Blaise Didry-Barca[2], Fanny Wondany[5], Tim Bergner[3], Tatiana Moreau[2], Mathieu P. Rodero[2], Reinhild Rösler[6], Sebastian Wiese[6], Stefano Volpi[7,8], Marco Gattorno[7], Riccardo Papa[7], Sally-Ann Lynch[9,10], Marte G. Haug[11], Gunnar Houge[12], Kristen M. Wigby[13,14], Jessica Sprague[15,16], Jerica Lenberg[14], Clarissa Read[3], Paul Walther[3], Jens Michaelis[5], Frank Kirchhoff[1], Carina C. de Oliveira Mann[4], Yanick J. Crow[2,17]✉ & Konstantin M. J. Sparrer[1]✉

Type I interferon (IFN) signalling is tightly controlled. Upon recognition of DNA by cyclic GMP-AMP synthase (cGAS), stimulator of interferon genes (STING) translocates along the endoplasmic reticulum (ER)-Golgi axis to induce IFN signalling. Termination is achieved through autophagic degradation or recycling of STING by retrograde Golgi-to-ER transport. Here, we identify the GTPase ADP-ribosylation factor 1 (ARF1) as a crucial negative regulator of cGAS-STING signalling. Heterozygous ARF1 missense mutations cause a previously unrecognized type I interferonopathy associated with enhanced IFN-stimulated gene expression. Disease-associated, GTPase-defective ARF1 increases cGAS-STING dependent type I IFN signalling in cell lines and primary patient cells. Mechanistically, mutated ARF1 perturbs mitochondrial morphology, causing cGAS activation by aberrant mitochondrial DNA release, and leads to accumulation of active STING at the Golgi/ERGIC due to defective retrograde transport. Our data show an unexpected dual role of ARF1 in maintaining cGAS-STING homeostasis, through promotion of mitochondrial integrity and STING recycling.

Type I interferon (IFN) mediated innate immunity constitutes an essential element in the response to viral infection, establishing an anti-viral state through the upregulation of hundreds of IFN stimulated genes (ISGs)[1–3]. Contrasting with this protective role against viral infection, inappropriate activation of an IFN response is pathogenic, eventually leading to tissue damage[4]. This dichotomy is particularly well illustrated by the type I interferonopathies, Mendelian diseases characterised by chronically enhanced type I IFN signalling[4,5]. Clinically, these disorders often manifest with neurological features such as

encephalopathy, cerebral calcification, leukodystrophy and cerebral atrophy. In addition, extra-neurological involvement can also be seen, with chilblain-like skin lesions a particularly frequent association. Notably, the study of the pathological basis of the type I interferonopathies has provided molecular insight into the induction and regulation of type I IFN signalling in human health and disease[4].

Type I IFNs can be induced through the sensing of foreign viral nucleic acids by innate immune receptors, among them cyclic GMP-AMP synthase (cGAS)[2,6–8]. Notably, cGAS can also be triggered by self-

---

DNA that is erroneously present in the cytoplasm[9]. Upon binding to cytoplasmic DNA, cGAS catalyses the production of a second messenger 2′−3′ cyclic guanosine monophosphate–adenosine monophosphate (cGAMP) from GTP and ATP[6,10]. cGAMP in turn binds to the signalling adaptor stimulator of IFN genes (STING), residing in its inactive form at the endoplasmic reticulum (ER), causing STING to change conformation, oligomerise, and accumulate in the Golgi[7]. Trafficking via the ER-Golgi axis is a major determinant of both STING activation and signal termination[11–13]. Activated STING is transported from the ER to the ER-Golgi intermediate compartment (ERGIC)/Golgi via coatomer protein complex II (COPII) vesicles. At the Golgi/ERGIC, STING recruits Tank binding kinase 1 (TBK1), which in turn activates the transcription factors IRF3 and NF-κB. Both transcription factors eventually translocate into the nucleus, leading to the induction of type I IFN and other (pro-)inflammatory cytokines. The current model proposes that, after TBK1 activation, STING is transported to the trans-Golgi network (TGN)/Golgi-associated vesicles where it is degraded via the autophagic-lysosomal pathway for signal termination[13]. In addition, increasing evidence suggests that STING can be recycled back to the ER[14–17]. Upon recruitment by Surfeit 4 (SURF4) to coatomer protein complex I (COPI) vesicles, STING is transported in a retrograde manner from the Golgi/ERGIC to the ER, thus also contributing to signal termination[15–17]. COPI trafficking is a highly conserved pathway down to yeast, functioning through the combined action of seven coatomer subunits: α-COP, β-COP, β′-COP, γ-COP, δ-COP, ε-COP, and ζ-COP. Despite these insights, the precise mechanism(s) and key players involved in STING flux and signal termination remain incompletely understood.

Small GTPases are important mediators of intracellular trafficking, among them the family of ADP-ribosylation factors (ARFs) comprising five highly homologous members in humans (ARF1, ARF2/4, ARF3, ARF5 and ARF6)[18]. These molecules have distinct but overlapping roles generally defined by their subcellular localisation[19]. For example, ARF1 is predominantly localised to the cis-Golgi, trans-Golgi and ERGIC[20], whereas ARF6 is present at the plasma membrane[21]. Their major function is to recruit coat proteins, such as COPI components, and mediate budding of transport vesicles from ER/Golgi surfaces. GTPase activity is required for vesicle formation and cargo recruitment by ARF proteins, cycling between a membrane-associated GTP-bound state and a GDP-bound cytoplasmic state[22]. GTPase activity, and GDP to GTP exchange, are regulated by a distinct set of GTPase accelerating proteins (GAP) and guanine nucleotide exchange factors (GEF), respectively, thereby in turn controlling the activity of ARF proteins[22].

Through the characterisation of a type I interferonopathy, we have discovered that the small GTPase ARF1 plays a key role in regulating cGAS-STING activity. ARF1 prevents aberrant type I IFN induction and signalling via a dual mechanism. First, our data suggests that ARF1 promotes mitochondrial dynamics, thus preventing aberrant release and sensing of mitochondrial DNA (mtDNA). In addition, ARF1 mediates retrograde transport of activated STING from the ERGIC/Golgi for signal termination. Mutation of the R99 residue in ARF1 is associated with a type I interferonopathy state, demonstrating attenuated GTPase activity and impaired function. Consequently, mtDNA leaks into the cytoplasm where it triggers cGAS. In addition, signal termination via ER-Golgi trafficking is defective, leading to elevated ISG expression. Our results reveal roles for ARF1 in cGAS-STING activation and signal termination essential to the maintenance of cellular homeostasis, and provide a mechanistic explanation for a previously unrecognized auto-inflammatory disease.

## Results

### Mutations in ARF1 define a type I interferonopathy

As part of an ongoing protocol involving the agnostic screening of patients with uncharacterized phenotypes for an upregulation of type I IFN signalling, we identified a patient (AGS460) with skin lesions and significant developmental delay to harbour a de novo c.295C>T/p.(R99C) substitution in ARF1 (Supplementary information). His skin disease was consistent with a diagnosis of chilblain lupus, a clinical sign frequently observed in a number of type I interferonopathies including Aicardi–Goutières syndrome (AGS) and STING-associated vasculopathy of infancy (SAVI) (Fig. 1a–c). Through Decipher and GeneMatcher/Matchmaker exchange[23], we ascertained two additional patients (RH2003, KW2022) with the same p.(R99C) substitution, in one of whom we could demonstrate that the variant arose de novo. Both of these patients exhibited skin lesions (Fig. 1d–g) similar to those observed in AGS460, in association with significant developmental delay. We then identified a further patient (AGS3133) with a de novo c.296G>A/p.(R99H) substitution and a history of cold hands and feet, without frank chilblains (Supplementary information).

ARF1 is a small GTPase with a myristylation anchor at the N-terminus that mediates membrane association[22,24]. This is followed by an amphipathic helix and the GTPase domain, consisting of two Switch domains (SW1 and 2) (Fig. 1h). Overall, ARF1 is highly constrained (pLi = 0.9). The arginine at position 99 is conserved from yeast to humans (Fig. 1i and Supplementary Fig. 1a), there are no variants at this residue on gnomAD, and a substitution for a cysteine or histidine is predicted as damaging by in silico analyses (Supplementary Data 1). Where tested, all (7 of 8) parents were found to be wild type (WT) on both alleles of ARF1 (Supplementary Fig. 1b). Of note, the p.(R99H) substitution was previously reported as a de novo mutation in a child with developmental delay and periventricular nodular heterotopia[25]. In AGS460 (Genotype ARF1 R99C), we observed increased ISG expression in peripheral blood on the two occasions assayed (at 14 and 17 years of age), and in AGS3133 (Genotype ARF1 R99H) on the one occasion tested at 15 years of age (Fig. 1j and Supplementary information). IFN signalling status could not be assessed in RH2003 or KW2022.

Taken together, these data indicate that mutation of the arginine at position 99 in ARF1 underlies a previously unrecognized human type I interferonopathy.

### ARF1 R99C induces STING-dependent type I IFN activation

To understand the molecular basis of enhanced type I IFN signalling in the context of this human type I interferonopathy, we examined the impact of ARF1 R99C expression on innate immune activation. The activity of ARF1 R99C was compared with ARF1 Q71L, which locks the protein in its GTP-bound state, and T31N which is trapped in the GDP-bound state[26]. In HEK293 cells stably expressing STING (293-Dual-hSTING-R232), transient expression of ARF1 R99C was sufficient to induce type I IFN signalling in a dose-dependent manner, as assessed by IFN stimulated response element (ISRE) reporter activity (Fig. 2a, b and Supplementary Fig. 2a). Of note, mutation of ARF1 R99 to H, A, E or K induced robust type I IFN signalling to a similar extent as R99C in reporter assays. This suggests that any substitution at position 99 would lead to chronic type I IFN release (Supplementary Fig. 2b, c). ARF1 Q71L also induced ISRE promoter activity, albeit less robustly than R99C, while ARF1 T31N had only a minor effect. STING was required for induction of type I IFN signalling by ARF1 R99C in HEK293T cells (Fig. 2c and Supplementary Fig. 2d). To avoid triggering high background cGAS-STING dependent responses by plasmid transfection, we then used a lentiviral transduction strategy. Lentivirus-mediated expression of ARF1 R99C in A549-ISRE reporter cells, that naturally express STING and cGAS, resulted in activation of type I IFN signalling (Fig. 2d). These data suggest that endogenous cGAS-STING proteins are sufficient for signalling induction. Expression of WT ARF1 had no effect, compared to treatment with IFN-β and cGAMP used as positive controls. Besides inducing type I IFN via IRF3, cGAS-STING signalling also enhances NF-κB activity[27]. In line with this, expression of ARF1 R99C in A549 cells induced NF-κB signalling (Fig. 2e), whereas expression of ARF1 WT had no effect. Activity of the kinase TBK1 is required downstream of cGAS-STING to activate NF-κB

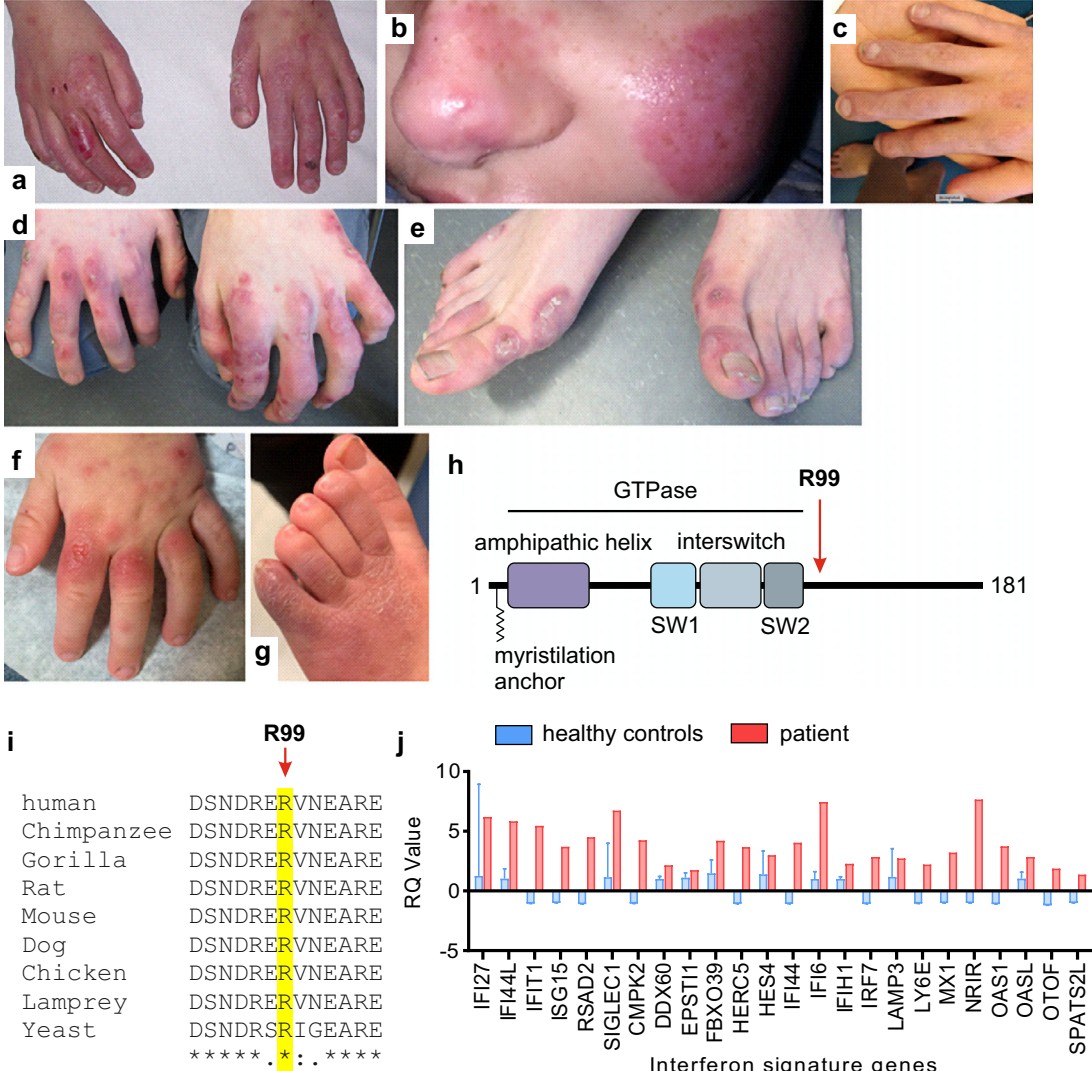

**Fig. 1 | Genetic identification of ARF1 R99C and clinical phenotype.**
**a**–**g** Erythematous vasculitic chilblain-like lesions on the hands (**a**) and face (**b**) of AGS460 aged 9 years. Right hand of the same patient at age 18 years showing frank tissue loss (**c**). Similar lesions were observed on the hands and feet of patients RH2003 (**d**, **e**) and KW2022 (**f**, **g**), at the ages of 18 and 2 years respectively. **h** Schematic overview of the domain structure of ARF1. Residue R99 is indicated by a red arrow. SW, Switch domain. **i** Multiple sequence alignment showing the conservation of residue R99 of ARF1 in indicated species. **j** ISG profile of patient AGS3133 (red) compared to the average of 27 healthy donors (blue). $n = 27$ (blue bars, healthy donors) ± SD; $n = 1$ (red bars, patient). See also Supplementary Fig. 1.

and IRF3[28]. As expected, overexpression of either R99C or Q71L ARF1 in HEK293T cells ectopically expressing cGAS and STING significantly increased the amount of active (i.e. phosphorylated) TBK1 (Fig. 2f, g). Of note, a decreased amount of endogenous STING protein upon co-transfection with ARF1 R99C is consistent with STING activation leading to degradation, as observed for STING activation by cGAMP stimulation (Fig. 2e)[13].

To assess these observations in a more physiological setting, we performed experiments using primary human fibroblasts derived from healthy individuals, and from an ARF1-mutated patient. First, primary human lung cells from healthy donors were complemented with ARF1 WT and ARF1 R99C by lentiviral transduction. Only the expression of ARF1 R99C, and exposure to the positive controls cGAMP and IFN-β, resulted in upregulation of ISG mRNAs (OAS1, Mx1) in primary human lung fibroblasts (Fig. 2h, Supplementary Fig. 2e). To determine whether patient-derived primary fibroblasts release type I IFN, we transferred supernatant from fibroblast cultures of healthy donors (n1 and f1, genotype ARF1 WT) and one patient (#AGS460, heterozygous ARF1 R99C) on to type I IFN signalling reporter HEK293T cells. Only

supernatants from the patient-derived cells resulted in a significant induction of the reporter (Fig. 2i).

Taken together, these results indicate that expression of ARF1 R99C induces a STING-dependent type I IFN response in cell line models and patient-derived primary cells.

## Mitochondrial DNA released in the presence of ARF1 R99C triggers cGAS activity

Since the induction of type I IFN signalling by ARF1 R99C was dependent on STING (Fig. 2c), we wondered whether and how cGAS may be involved. Inhibition of cGAS activity using a pharmacological inhibitor, G140, significantly reduced ISRE promoter activation by ARF1 R99C in 293-Dual-hSTING-R232 cells (Supplementary Fig. 3a). In line with this, expression of ARF1 WT and R99C via lentiviral transduction in WT and cGAS KO THP-1 monocytes revealed that IFN signalling was only induced in the presence of cGAS (Fig. 3a). As expected, IFN signalling upon stimulation with IFN-β and cGAMP was barely affected. These results suggest that cGAS facilitates type I IFN signalling induced by ARF1 R99C. ARF1 has been reported to regulate mitochondrial

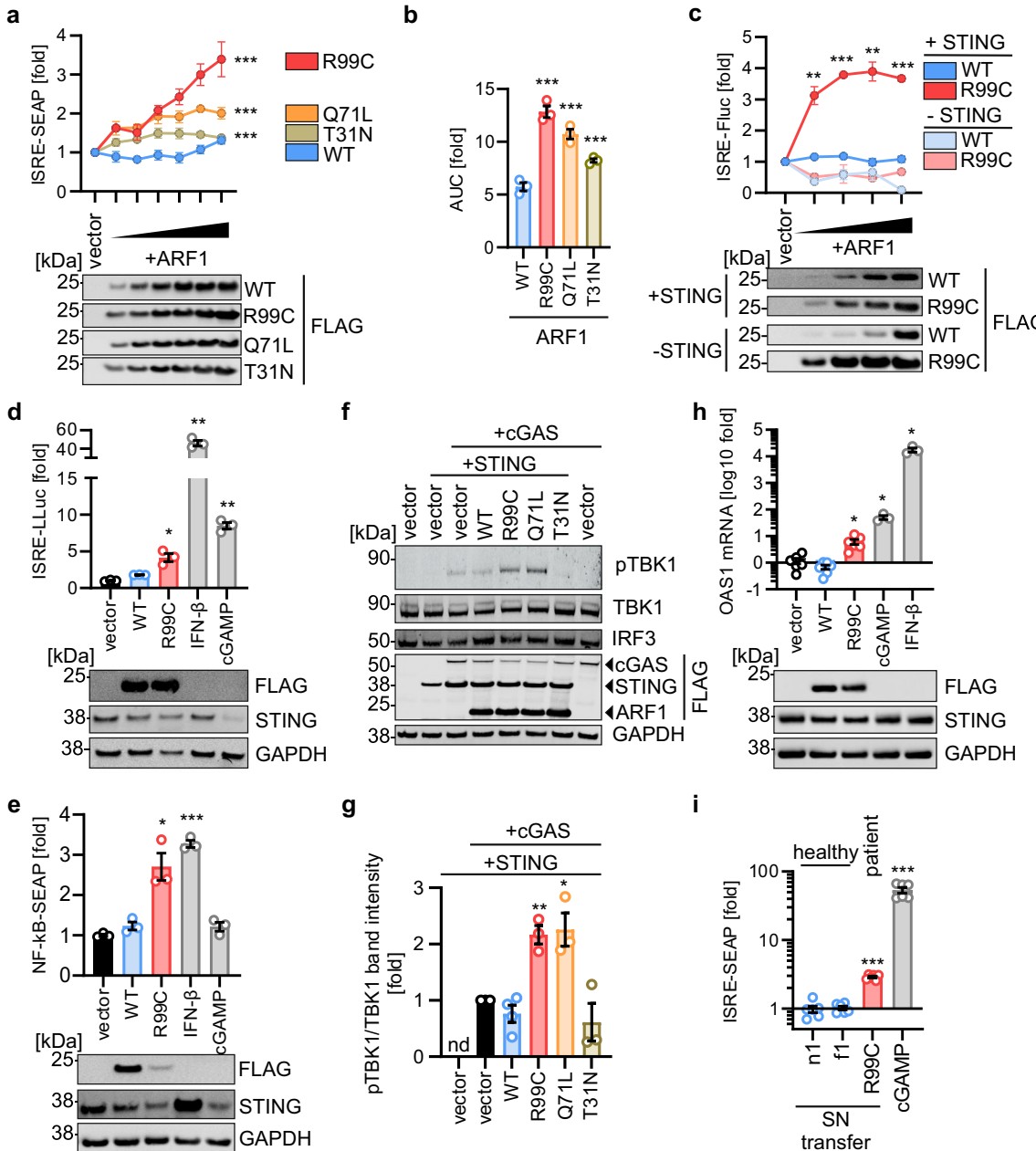

**Fig. 2 | Expression of ARF1 R99C induces a STING-dependent type I IFN response. a** ISRE promoter activity by SEAP reporter in 293-Dual-hSTING-R232 cells transiently expressing FLAG-tagged ARF1 WT, R99C, Q71L or T31N on (32 h post transfection and normalised to cell viability). Dots represent mean of $n = 3 \pm$ SEM (biological replicates). Lower panel: Corresponding immunoblots of whole-cell lysates (WCLs) stained with anti-FLAG. **b** Area under the curve analysis of the data in (**a**). **c** ISRE promoter activity by Firefly luciferase (Fluc) quantification in HEK293T cells transiently expressing STING (+STING) or empty vector (-STING) and FLAG-tagged ARF1 WT and R99C (32 h post transfection and normalised to GAPDH-Renilla luciferase). Dots represent mean of $n = 3 \pm$ SEM (biological replicates). Lower panel: Corresponding immunoblots of WCLs stained by anti-FLAG. **d, e** ISRE (**d**) or NF-κB (**e**) promoter activity in A549 Dual cells transduced with ARF1 WT and R99C. IFN-β (1000 U/mL, 16 h) and cGAMP (10 μg/ml, 16 h) served as positive controls. Bars represent mean of $n = 3 \pm$ SEM (biological replicates). Lower panels: Corresponding immunoblots of WCLs stained by anti-FLAG, anti-STING and anti-GAPDH. **f** Exemplary immunoblot of HEK293T WCLs transiently expressing indicated ARF1 constructs, cGAS and STING. Blots were stained with anti-pTBK1, anti-TBK1, anti-IRF3, anti-FLAG and anti-GAPDH. **g** Quantification of the pTBK1 band intensities in (**f**) normalized to TBK1. Bars represent mean of $n = 4$ (vector, WT), $n = 3$ (R99C, Q71L, T31N) $\pm$ SEM (biological replicates). **h** qPCR of OAS1 mRNA in primary normal human lung fibroblasts transduced to express ARF1 WT and R99C 72 h post transduction. IFN-β (1000 U/mL, 16 h) and cGAMP (10 μg/ml, 16 h). Bars represent mean of $n = 3$ (cGAMP, IFN-β) and $n = 6$ (vector, WT, R99C) $\pm$ SEM (biological replicates). Lower panel: Corresponding immunoblots of WCLs stained with anti-FLAG, anti-STING and anti-GAPDH. **i** Supernatant (SN) transfer of primary fibroblasts from healthy donors (n1, f1) or a patient (AGS460) to 293-Dual-hSTING-R232 cells. IFN-β (100 U/mL, 48 h) and cGAMP (10 μg/ml, 48 h). SEAP (ISRE) activity 48 h post transfer and normalised to cell viability. Bars represent mean of $n = 5$ (n1) and $n = 6$ (f1, R99C, cGAMP) $\pm$ SEM (biological replicates). See also Supplementary Fig. 2. Statistical analysis was performed using two-tailed Student's $t$ test with Welch's correction. *$p < 0.05$; **$p < 0.01$; ***$p < 0.001$. Exact $P$ values and Source data are provided in the Source data file.

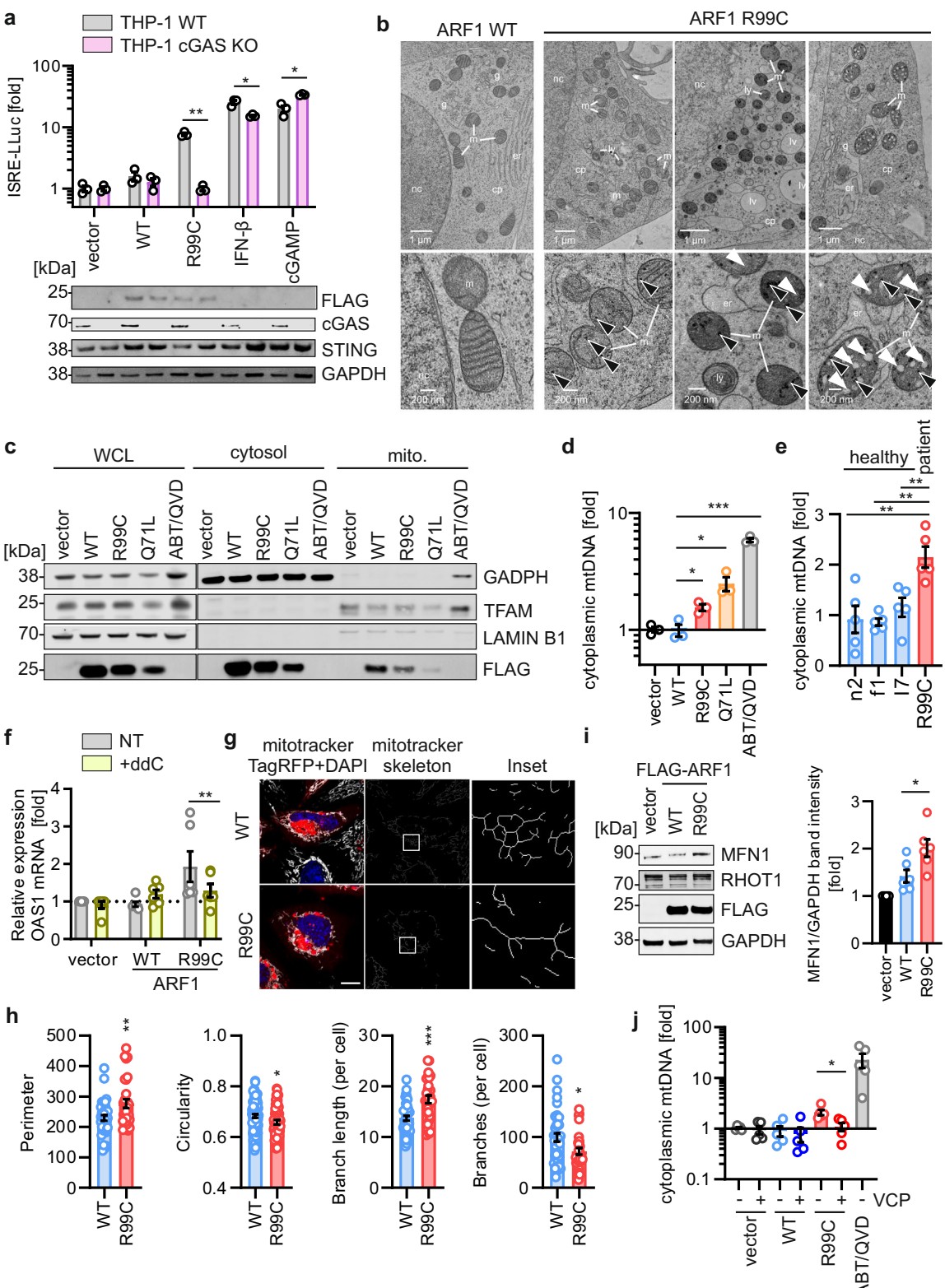

homeostasis[29,30]. Thus, we examined whether mutation at R99 may disrupt this function, promoting release of mtDNA, which in turn might activate cGAS[31,32]. Electron microscopy of HEK293T cells revealed mitochondrial disruption in the presence of ARF1 R99C, but not WT ARF1 (Fig. 3b). Indicative of mitochondrial damage, small electron-dense granules were observed in the mitochondria of R99C-expressing cells (Fig. 3b, left panel). With increasing mitochondrial degeneration these granules became larger, and the intermembrane

space of the cristae more inflated (Fig. 3b, middle and right panel). To determine the presence of mtDNA in the cytoplasm, we performed fractionation of cells into nuclear, cytoplasmic and mitochondrial fractions (Fig. 3c). We detected higher levels of mtDNA in the cytoplasm upon overexpression of ARF1 R99C and Q71L compared to WT ARF1 (Fig. 3d). Treatment with ABT-737 and Q-VD-OPH (ABT/QVD), to damage mitochondria and promote mtDNA release[33], was used as a positive control. Notably, in primary fibroblasts of a patient with a

**Fig. 3 | ARF1 R99C disrupts mitochondria, releasing mtDNA into the cytoplasm.**
**a** ISRE promoter activity quantified by Lucia luciferase (LLuc) in THP-1-Dual WT (grey) or cGAS KO (pink) transduced to express ARF1 WT or R99C (72 h post transduction). IFN-β (1000 U/mL, 16 h) and cGAMP (10 μg/ml, 16 h). Bars represent mean of $n = 3 \pm$ SEM (biological replicates). Lower panel: Corresponding immunoblots of WCLs stained by anti-FLAG, anti-STING, anti-cGAS and anti-GAPDH.
**b** Exemplary electron microscopy analysis of HEK293T cells transiently expressing ARF1 WT or R99C. Electron-dense granules, black arrows. Inflated cristae, white arrows. cp, cytoplasm. er, endoplasmic reticulum. g, Golgi apparatus. lv, large vesicle. ly, lysosome. m, mitochondria. nc, nucleus. **c** Exemplary immunoblots showing fractionation of HEK293T cells transiently expressing ARF1 WT, R99C, Q71L or vector. WCLs and fraction blots stained with anti-FLAG, anti-TFAM, anti-LAMIN B1 and anti-GAPDH. This experiment was repeated to similar results independently at least two times. **d** qPCR of mtDNA (MT-D-Loop) in the cytosolic fraction of (**c**) normalized to cellular mtDNA (mtDNA/nuclear DNA). $n = 3 \pm$ SEM. **e** qPCR of mtDNA (MT-D-Loop) in the cytosolic fraction of primary fibroblasts of healthy donors (n2, f1, I7) or patient (AGS460) relative to total normalized cellular mtDNA (mtDNA/nuclear DNA). $n = 5 \pm$ SEM. **f** qPCR of *OAS1* mRNA in U2OS cells stably expressing STING and depleted of mtDNA by ddC, or untreated (NT).

Transiently transfected with empty vector, ARF1 WT or R99C. $n = 6 \pm$ SEM.
**g** Exemplary live cell confocal laser scanning microscopy images of HeLa cells transiently expressing TagRFP-tagged ARF1 WT, R99C or vector control (red). Cells were stained with Mitotracker (grey, 1 μM, 30 min, 37 °C). Nuclei, Hoechst (blue). Scale bar, 10 μm. **h** Analysis of the perimeter, circularity, branch length (per cell) and branches (per cell) of the images in (**g**). Lines represent mean of $n = 37$ (WT, perimeter), $n = 29$ (R99C, perimeter), $n = 80$ (WT, circularity), $n = 60$ (R99C, circularity), $n = 40$ (WT, branch length, branches), $n = 29$ (R99C, branch length, branches) $\pm$ SEM. **i** Exemplary immunoblot of WCLs of HEK293T cells transiently expressing ARF1 WT, R99C or vector. Blots were stained with anti-MFN1, anti-RHOT1, anti-FLAG and anti-GAPDH. Quantification of the band intensities of MFN1 normalized to GAPDH. Bars represent mean of $n = 6 \pm$ SEM (biological replicates). **j** qPCR of mtDNA (MT-D-Loop) in the cytosolic fraction of HEK293T cells transiently expressing ARF1 WT, R99C or vector control as well as VCP normalized to cellular mtDNA (mtDNA/nuclear DNA). $n = 5 \pm$ SEM. See also Supplementary Fig. 3. Statistical analysis was performed using two-tailed Student's $t$ test with Welch's correction or two-way ANOVA (**f**). $*p < 0.05$; $**p < 0.01$; $***p < 0.001$. Exact P values and Source data are provided in the Source data file.

heterozygous R99C mutation in ARF1 (AGS460), cytoplasmic mtDNA levels were significantly elevated compared to control ARF1 WT donor fibroblasts (n2, f1, I7) (Fig. 3e). Depletion of mtDNA using 2′,3′ dideoxycytidine (ddC)[34] in U2OS-STING cells reduced the induction of the ISG OAS1 in the presence of ARF1 R99C (Fig. 3f, Supplementary Fig. 3b). In line with this, expression of *IFNB1*, *Mx1* and *RSAD2* induced by ARF1 R99C expression was also reduced upon mtDNA depletion (Supplementary Fig. 3d–f). Of note, mtDNA depletion did not change the response to herring testis DNA (HT-DNA) (Supplementary Fig. 3c).

In summary, these data show that expression of ARF1 R99C promotes mitochondrial disruption and leakage of mtDNA into the cytoplasm, and a subsequent induction of ISG expression.

### ARF1 R99C destabilizes mitochondria by interfering with mitofusin 1 levels

Since ARF1 has been reported to be involved in mitophagy (i.e. autophagic turnover of defective mitochondria), and in the regulation of mitochondrial fusion and fission[29,35], an impairment of either process by ARF1 R99C might cause aberrant mtDNA release via disturbed mitochondrial homeostasis[36–38]. Thus, we explored whether mitophagy is altered by ARF1 R99C, with subsequent impact on mtDNA release. In the presence of both ARF1 WT and R99C, autophagosomes accumulated in HEK293T GFP-LC3B reporter cells (Supplementary Fig. 3g). Bafilomycin A1 treatment masked the effect of both ARF1 WT and R99C, suggesting that both proteins reduce autophagosome turnover (Supplementary Fig. 3h). Notably, in autophagy-impaired ATG5 KO cells, ARF1 R99C still mediated mtDNA release compared to WT and vector conditions (Supplementary Fig. 3i), suggesting that a defect in autophagy is not responsible for mtDNA release in the presence of ARF1 R99C. Mitotracker staining of mitochondria in transfected HeLa cells revealed a significant reduction in the mitochondrial footprint in the presence of ARF1 R99C (Supplementary Fig. 3j, k), indicating changes in mitochondrial fission or fusion. In depth analysis of the mitochondrial network stained by mitotracker in the presence of WT or R99C ARF1 in HeLa cells revealed that the mitochondrial perimeter is significantly increased, whereas the circularity of the mitochondria is reduced (Fig. 3g, h). Furthermore, quantification of the mitochondrial skeleton network showed that in the presence of ARF1 R99C the branch length is increased, and, conversely, the number of branches in the network per cell reduced (Fig. 3g, h). This suggests that the mitochondria are elongated, typically expected either upon decreased fission or increased fusion[39,40]. Subsequent analysis of phosphorylated levels of the mitochondrial fission factor Drp1 revealed no change in the presence of ARF1 R99C (Supplementary Fig. 3l). In contrast, the mitochondrial fusion marker MFN1

accumulated significantly in the presence of ARF1 R99C but not ARF1 WT (Fig. 3i). This suggests that increased fusion but not alterations in fission explain the perturbation of mitochondria in the presence of ARF1 R99C. Consistently, ER-mitochondria contacts, that occur during mitochondrial fusion, were visibly increased (Fig. 3b). In yeast, ARF1 deficiency leads to Fzo1 (yeast equivalent of MFN1/2) misfolding and aggregation[29]. To understand whether this impacts mtDNA release by ARF1 R99C, we induced MFN1 turnover by overexpression of the E3 ubiquitin ligase Valosin Containing Protein (VCP)[29]. Notably, overexpression of VCP reduced mtDNA release by ARF1 R99C almost back to basal levels (Fig. 3j, Supplementary Fig. 3m).

Taken together, these data suggest that ARF1 R99C promotes hyperfusion of mitochondria by MFN1 accumulation, leading to a leakage of mtDNA into the cytoplasm.

### Mutation of R99 impairs ARF1 GTPase activity and association with the COPI complex

ARF1 is a small GTPase, and both GTP-GDP cycling and co-factor binding are known to be crucial for its enzymatic activity[18]. Analyses of ARF1 structures, based on PDB: 2J59[41], indicated that residue R99 is not directly involved in binding to GTP or other ARF1 interaction partners. Instead, R99, located on helix α5, forms salt bridges with residue D26, thereby stabilizing the loop β1-α3 which is part of the GTP-binding site of ARF1 (Fig. 4a). D26 is not directly involved in the coordination of GTP binding (Fig. 4a). However, correct positioning of the loop β1-α3 is crucial for the stability of ARF1 and GTP hydrolysis[41]. To experimentally assess the effects of the R99C mutation on protein stability, GTP binding and GTP hydrolysis, we purified recombinant ARF1 WT, ARF1 R99C and ARF1 Q71L (Supplementary Fig. 4a), and performed comparative fluorescence thermal shift assays as well as in vitro GTPase assays. Both ARF1 WT and R99C proteins bind GTP with similar affinity, as revealed by an increase of the melting temperatures by -16 °C of both proteins in the presence of GTP (Fig. 4b, c). However, a 15 °C lower melting temperature of ARF1 R99C compared to ARF1 WT or ARF1 Q71L, indicates reduced conformational stability (Supplementary Fig. 4b). Consequently, the in vitro GTPase activity of ARF1 R99C was significantly lower than that of WT ARF1 (Fig. 4d). Of note, reduced conformational stability does not lead to low expression levels of ectopically expressed R99C ARF1, which would be indicative of an unstable protein (Figs. 2–4). In addition, ARF1 dimerization, known to be required for vesicle transport activity and GTP hydrolysis[18,42], was similar between mutant and WT ARF1 (Supplementary Fig. 4c). To promote the GTPase activity of ARF1 R99C, we co-expressed mutant ARF1 with the ARF1 GTPase activating protein (GAP) ARFGAP1, and with the GTP exchange factor (GEF) BIG1[22]. Accelerating GTPase

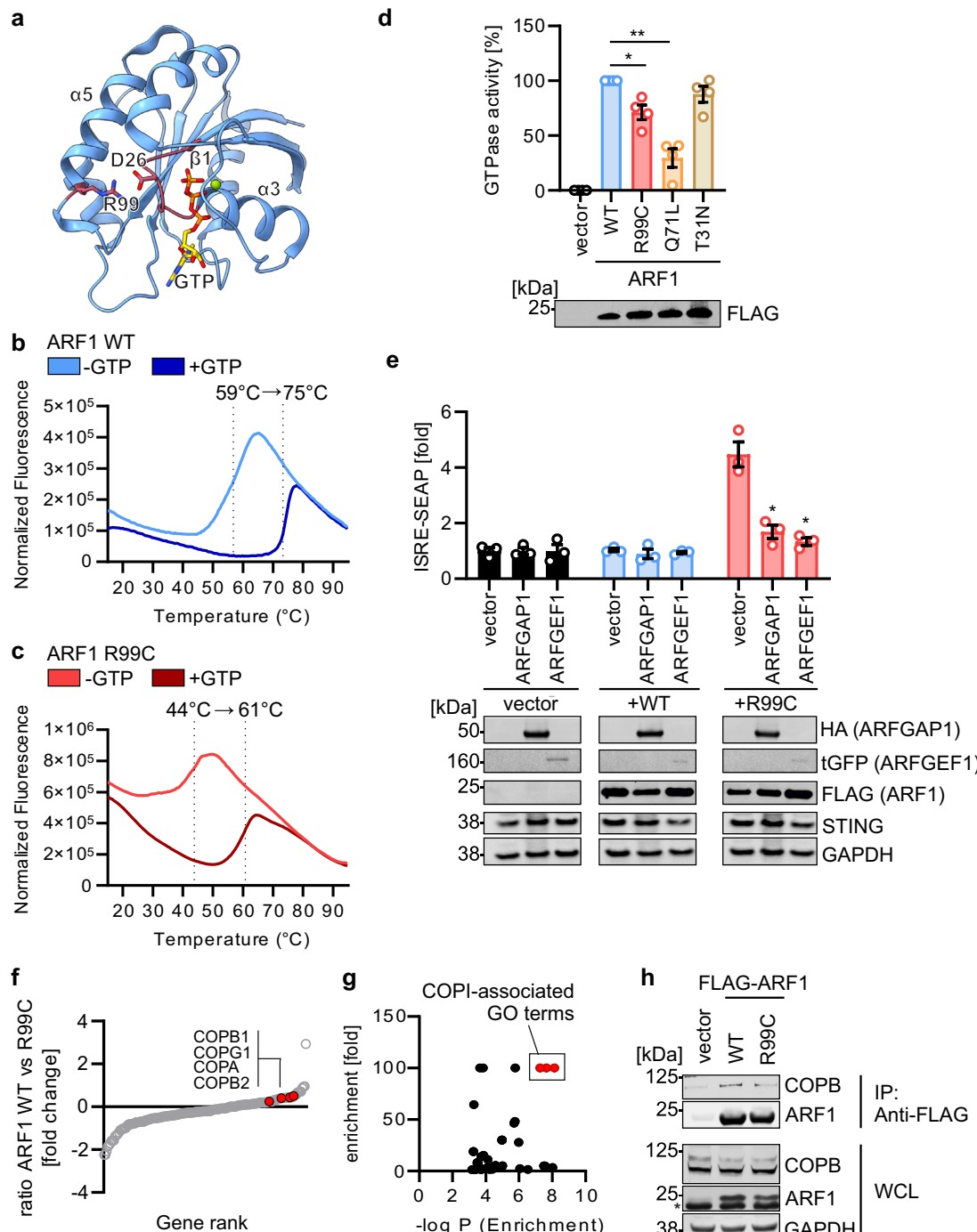

**Fig. 4 | GTPase activity of ARF1 R99C is reduced. a** Model of ARF1 (PDB: 2J59) ATP-bound (orange/yellow). R99 and D26 are highlighted. **b**, **c** Interaction of ARF1 WT (**b**) or R99C (**c**) with GTP and protein stability analysed by fluorescence thermal shift assay. Data are representative of two biological replicates. **d** GTPase activity of indicated ARF1 mutants purified from HEK293T cells expressing FLAG-ARF1. Lower panel: Corresponding immunoblot stained with anti-FLAG. Bars represent mean of $n = 4 \pm$ SEM (biological replicates). **e** ISRE promoter activity in 293-Dual-hSTING-R232 cells quantified by SEAP transiently expressing ARF1 WT or R99C with indicated GEF or GAP normalized to cell viability. Corresponding immunoblots of WCLs stained with anti-HA, anti-turboGFP (tGFP), anti-FLAG, anti-STING and anti-GAPDH. Bars represent mean of $n = 3 \pm$ SEM (biological replicates). **f** Protein abundance in ARF1 WT versus R99C large scale purification from HEK293T cells as assessed by SILAC mass spectrometry. Components of the COPI machinery are highlighted in red. **g** Gene Ontology Analysis (PantherDB) of the top 30 proteins less associated with ARF1 R99C in (**f**). Fold enrichment of individual GO terms versus the $-\log P$ value. COPI-associated GO terms are highlighted in red. **h** Co-immunoprecipitation using anti-FLAG beads from WCLs of HEK293T cells transiently expressing FLAG-ARF1 WT, R99C or vector. Immunoblots stained with anti-COPB, anti-ARF1 and anti-GAPDH. Asterisk denotes unspecific background band. The experiment was repeated three times independently to similar results. See also Supplementary Fig. 4. Statistical analysis was performed using two-tailed Student's $t$ test with Welch's correction. *$p < 0.05$; **$p < 0.01$; ***$p < 0.001$. Exact $P$ values and Source data are provided in the Source data file.

activity by a GAP, or increasing GDP/GTP exchange by a GEF, rescued aberrant type I IFN induction by ARF1 R99C in 293-Dual-hSTING-R232 cells (Fig. 4e).

Destabilization of ARF1 may also result in decreased binding to important effectors and co-factors. To explore this possibility, we performed stable isotope labelling of amino acids in cell culture (SILAC) mass spectrometry experiments, comparing the interaction of endogenous proteins to either ARF1 WT or ARF1 R99C expressed in HEK293T cells, in heavy and light arginine containing media, respectively (Supplementary Fig. 4d). Proteins were purified while maintaining endogenous interaction partners (Supplementary Fig. 4e), and ~1700 co-purified proteins were identified by mass spectrometry analysis of the lysates of two independent pulldowns (Supplementary Data 2). The ratio between ARF1 (heavy labelled) and ARF1 R99C (light labelled) was at 1 and their intensity the highest among the detected proteins, suggesting similar pulldown efficiency (Supplementary Fig. 4f). For further analysis, the 166 proteins that were detected in both independent replicates were considered (Fig. 4f and Supplementary Fig. 4f). This analysis suggested that major protein components of COPI complex, COPA, COPB1, COPB2, COPG1, showed 1.18, 1.32, 1.36 or 1.41-fold lower association with ARF1 R99C. Panther DB aided Gene Ontology (GO) term analysis of the top 30 proteins binding with lower affinity to ARF1 R99C showed that interaction with proteins of the COPI pathway was significantly reduced by the mutation R99C (GoTerm: COPI vesicle coat (GO:0030126), >100-fold enrichment in analysed list, $p = 7.42E\text{-}09$; COPI-coated vesicle membrane (GO:0030663), >100-fold enrichment in analysed list, $p = 2.27E\text{-}08$; COPI-coated vesicle (GO:0030137), 100-fold enrichment in analysed list, $p = 5.42E\text{-}08$) (Fig. 4g, Supplementary Data 3). All human genes in the PantherDB database were used as a reference list. In line with the SILAC experiments, COPB associated less to overexpressed ARF1 R99C in HEK293T cells than ARF1 WT in co-immunoprecipitation experiments (Fig. 4h).

Altogether, biochemical characterization revealed that R99C reduced conformational stability, GTPase activity and association of ARF1 with components of the COPI complex.

## ARF1 is required for retrograde transport from the ERGIC to the ER

To understand whether the reduced association of ARF1 R99C with components of the COPI transport machinery might affect STING trafficking and signalling directly, we sought to activate STING independently of cGAS. Indeed, bypassing the requirement for cGAS by treatment with cGAMP showed that ARF1 R99C further enhances IFN signalling compared to WT ARF1 (Fig. 5a, b). Consistently, we observed elevated IFN signalling in the presence of ARF1 R99C upon expression of an active STING mutant (R238A/Y240A) that does not bind or require cGAMP[43,44] (Fig. 5c and Supplementary Fig. 5a). As expected, STING with a mutation in S366A, that is unable to bind to TBK1, did not induce type I IFN signalling, despite being expressed at similar levels to WT STING (Supplementary Fig. 5b). Neither ARF1 WT nor R99C caused further signalling activation. The above results suggest that, in addition to promoting mtDNA leakage to the cytoplasm, ARF1 R99C also enhances type I IFN signalling through changes in STING trafficking and signalling downstream of cGAS/cGAMP. As STING recycling has been proposed to occur by retrograde transport via COPI vesicles[14–17], we then asked if the decreased association of ARF1 R99C with COPI vesicles might result in decreased STING recycling, and thus impaired signal termination of STING and prolonged IFN induction. To this end, we explored the impact of patient-associated mutant ARF1 on retrograde Golgi/ERGIC-ER transport. Electron microscopy analysis revealed that ARF1 R99C expression led to altered Golgi/ERGIC structures, whereas empty vector or ARF1 WT transfection had no effect (Fig. 5d). Indeed, scanning-transmission-electron-microscopy

(STEM) tomography analysis revealed that, in the presence of ARF1 R99C, the volume density of the lumen of the Golgi is increased (Fig. 5e), whereas the volume density of vesicles released from the Golgi and the ER is significantly reduced compared to ARF1 WT (Fig. 5f). Focusing on the ERGIC, confocal microscopy analysis confirmed the alteration of Golgi structure. Specifically, it showed that the number of ERGIC-53 positive vesicles per cell i.e. ERGIC vesicles, was significantly reduced in the presence of ARF1 R99C and ARF1 Q71L (Fig. 5g, left panel, Supplementary Fig. 5c). Conversely, the area occupied by individual ERGIC-53 positive structures was increased upon ARF1 R99C and Q71L expression (Fig. 5g, middle panel), while the total area of the ERGIC-53 positive structures per cell did not change markedly (Fig. 5g, right panel). Thus, in the presence of ARF1 R99C, fewer but larger vesicles/ERGIC structures are present, suggesting impaired budding of vesicles from the ERGIC. To examine whether ARF1 R99C impacts general ERGIC to ER trafficking, we used a thermolabile VSV-G protein fused to the KDEL retrograde trafficking signal[45]. Upon decreasing the temperature to 32 °C, the normally ER-resident construct is transported to the Golgi, while shifting to 40 °C partially unfolds VSV-G thereby inducing retrograde transport back to the ER. Both ARF1 WT and R99C expression did not impact VSV-G increased co-localisation with the cis-Golgi (marker GM130) upon lowering the temperature (Fig. 5h, Supplementary Fig. 5d). However, when raising the temperature to 40 °C, retrograde transport was almost completely abrogated in the presence of ARF1 R99C, but unaffected by ARF1 WT expression (Fig. 5h).

In summary, the above data demonstrate that ARF1 R99C causes a defect in Golgi-ER retrograde trafficking and reduced vesicular budding from the Golgi/ERGIC.

## Active STING is trapped at the Golgi/ERGIC by ARF1 R99C

To examine the consequences of defective retrograde transport on STING in the presence of ARF1 R99C, we examined the co-localisation of ARF1 and STING-eGFP in HeLa cells relative to endogenous markers of the Golgi network (Fig. 6a–f, Supplementary Fig. 6a–c). As controls, ARF1 Q71L is expected to show increased membrane association with the Golgi/ERGIC, whereas ARF1 T31N is thought to be more cytoplasmic[18]. While ARF1 WT, R99C and Q71L localised similarly to the cis-Golgi as revealed by co-localisation with GM130, ARF1 T31N showed a reduced co-localisation with GM130 (Fig. 6a, b). Notably, expression of both ARF1 R99C and Q71L led to a significant increase of eGFP-STING localization at the cis-Golgi signal compared to WT and T31N ARF1 (Fig. 6a, c). Analysis of ERGIC localisation revealed that ARF1 R99C and Q71L showed markedly increased co-localisation with the ERGIC compared to ARF1 WT (Supplementary Fig. 6a–c). Consistent with this, expression of ARF1 R99C and Q71L significantly increased the co-localisation between eGFP-STING and ERGIC-53, whereas ARF1 WT and T31N had little or no impact on STING localisation at the ERGIC (Supplementary Fig. 6a–c). The presence of ARF1 R99C did not alter the localisation of eGFP-STING with respect to the trans-Golgi network (TGN46) (Supplementary Fig. 6d–f). In primary fibroblasts from a healthy donor (genotype ARF1 WT), endogenous STING only accumulates at the Golgi network (GM130) upon stimulation with cGAMP (Fig. 6d, e). In contrast, in fibroblasts from a patient heterozygous for the R99C mutation in ARF1 (#AGS460), STING accumulated at the cis-Golgi/ERGIC even in the absence of cGAMP stimulation. This is consistent with the data from our cell line experiments (Fig. 6d, e). To analyse the effect of ARF1 mutants on the localisation of endogenous STING in more detail, we employed super-resolution microscopy (stimulated emission/depletion microscopy, STED). Two colour STED microscopy in primary human lung fibroblasts transduced with empty, ARF1 WT or ARF R99C expressing lentiviruses confirmed that, in the presence of ARF1 R99C, endogenous STING accumulates at/around the cis-Golgi (GM130) network (exemplary image in Fig. 6f, further images in Supplementary Fig. 6g).

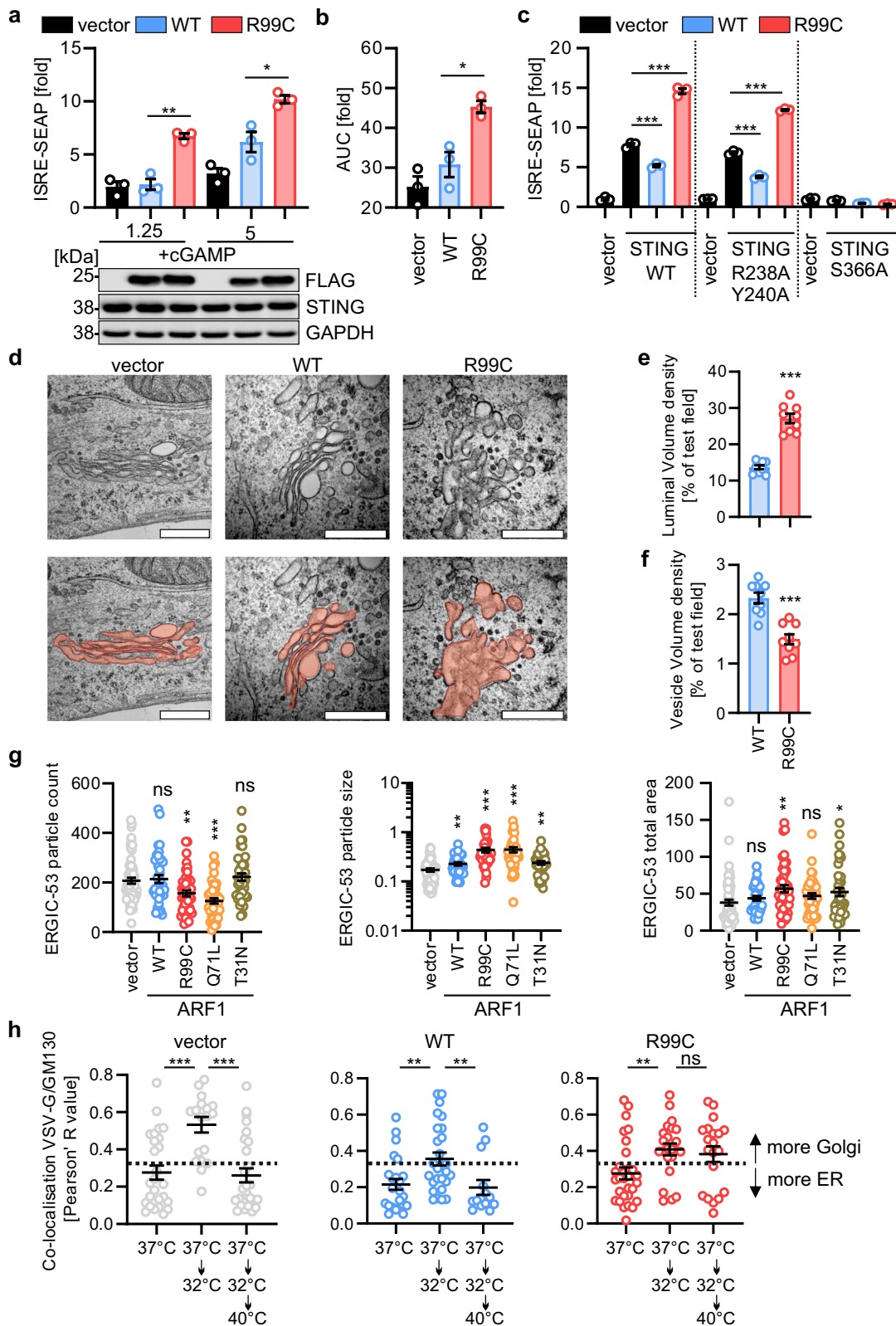

To determine whether STING accumulates in its active state at the Golgi/ERGIC, we co-stained phospho-TBK1 (p-S172) as a marker of STING activity in situ. Confocal analysis revealed that in the presence of transduced ARF1 R99C, but not ARF1 WT, increased amounts of activated TBK1 were present in the proximity of the ERGIC in primary human lung fibroblasts (Fig. 6g, h).

Taken together, these results suggest that ARF1 plays a role in the retrograde transport of STING from the Golgi/ERGIC to the ER. When trafficking is affected due to impaired ARF1 GTPase activity (e.g. by mutations at R99C or Q71L), STING accumulates at the cis-Golgi/ERGIC in both cell models and primary cells isolated from patients, leading to activation of TBK1 and sustained IFN induction/signalling.

**Fig. 5 | ARF1 is responsible for retrograde transport from the Golgi/ERGIC to the ER. a** ISRE promoter activity quantified by SEAP in 293-Dual-hSTING-R232 cells transiently expressing FLAG-tagged ARF1 WT or R99C (32 h post transfection, normalized to cell viability). cGAMP (1.25 μg/ml or 5 μg/ml, 16 h). Bars represent mean of $n = 3 \pm$ SEM (biological replicates). Lower panel: Representative corresponding immunoblots stained with anti-FLAG, anti-STING and anti-GAPDH. **b** Area under the curve analysis of the data in (**a**). **c** ISRE-promoter activity quantified by firefly luciferase in HEK93T cells transiently expressing ARF1 WT or R99C in the presence of indicated STING mutants (32 h post transfection, normalised to GAPDH-Renilla luciferase). Bars represent mean of $n = 3 \pm$ SEM (biological replicates). **d** Representative electron microscopic images of HEK293T cells transiently expressing indicated ARF1 mutants. Luminal area (orange). Scale bar, 0.5 μm. **e** Golgi/ERGIC luminal volume and **f** associated vesicle volume quantified in tomograms of cells in (**d**), as percentage of the total volume of the analysed section.

Bars represent mean of $n = 9 \pm$ SEM (images). **g** Quantification of particles of ERGIC-53 staining (left panel), particle size (middle panel) or the total are (right panel), in HeLa cells transiently expressing ARF1 WT, R99C, Q71L or T31N. Cells were stained 24 h post transfection with anti-ERGIC-53 and anti-FLAG. Black lines represent mean of $n = 61$ (vector), $n = 39$ (WT), $n = 45$ (R99C), $n = 41$ (Q71L, T31N) $\pm$ SEM (individual cells). **h** Pearson's correlation coefficient indicating co-localisation between GM130 and thermosensitive VSV-G (VSVG-ts045) in HeLa cells transiently expressing VSVG-ts045-KDELR, ARF1 WT or ARF1 R99C. Temperature shifts (37 °C/32 °C/40 °C) as indicated. Lines represent mean of $n = 27/19/27$ (vector, 37 °C/32 °C/40 °C), $n = 24/27/14$ (WT, 37 °C/32 °C/40 °C), $n = 29/23/18$ (R99C, 37 °C/32 °C/40 °C) $\pm$ SEM (cells). See also Supplementary Fig. 5. Statistical analysis was performed using two-tailed Student's $t$ test with Welch's correction.: *$p < 0.05$; **$p < 0.01$; ***$p < 0.001$. Exact $P$ values and Source data are provided in the Source data file.

## Discussion

The avoidance of chronic activation of a type I IFN response is fundamental to immunological homeostasis[4]. Engagement of the cGAS-STING pathway occurs through the recognition of DNA by cGAS ('Trigger'), and subsequent trafficking of STING from the ER to the Golgi. To terminate such signalling, STING is transported back to the ER ('Recycling') and degraded via autophagy ('Removal'). Through an analysis of the function of an ARF1 mutant associated with elevated IFN signalling in vivo, we have identified a previously unrecognised dual role of the GTPase ARF1 in both preventing aberrant cGAS activation and promoting signal termination by the recycling of STING (Supplementary Fig 7). We show that functional ARF1 promotes mitochondrial integrity, thereby preventing aberrant stimulation of cGAS by mtDNA. In addition, we demonstrate a role for ARF1 in the termination of cGAS-STING signalling, by facilitating COPI vesicle-mediated trafficking of STING from the ERGIC to the ER. When the GTPase activity of ARF1 is impaired, either by disease-associated mutations at R99 or by the characterised GTP-locked Q71L mutation, type I IFN responses are enhanced. Consequently, patients bearing a heterozygous R99 mutation in ARF1 manifest features of chronic innate immune activation.

Complete disruption (e.g. by drugs such as Brefeldin A) or depletion of ARF1 is well known to decrease STING-dependent signalling[11,12,46]. This effect is most likely due to a generalised disturbance of the Golgi structure/transporting system[47], including the anterograde transport of STING required for its activation. In contrast, our investigation of a physiologically relevant human interferonopathy-associated ARF1 mutant has allowed us to define an ARF1-specific effect on retrograde STING trafficking, thus differentiating it from a non-specific effect on anterograde transport.

GTPases are core components involved in the regulation of mitochondrial fission and fusion that include MFN1, MFN2, and OPA1. A role of ARF1 in regulating Fzo1 (the yeast MFN1 homologue) was previously proposed in yeast[29]. In the absence of functional ARF1, Fzo1 accumulated, leading to a loss of mitochondrial fusion. Our data show that human ARF1 also regulates mitochondrial fusion. Thus, non-functional ARF1 leads to impaired mitochondrial integrity and aberrant accumulation of MFN1. The human homologue of yeast Fzo1 associated E3 ubiquitin ligase Cdc48, VCP, was able to resolve the mitochondrial fusion impairment consequent upon defective ARF1. These data suggest that ARF1 plays an important role in mitochondrial fusion and fission in mammals. Of note, it was recently shown that Golgi-derived vesicles may aid mitochondrial fusion and fission, spatially linking ARF1 and mitochondria[30,48]. However, the precise role of ARF1 in mitochondrial fusion will require future study.

Previous work has shown that defects in retrograde transport can mediate chronic type I IFN release in a cGAS dependent[17] or independent fashion[14]. Our mechanistic analyses indicate that IFN induction in patients with ARF1 R99C is dependent on cGAS activity. Given impaired mitochondrial integrity in both model and patient cells, our data suggest that, even in the absence of a pathogen-derived cGAS

trigger, ARF1 dysfunction may lead to sterile inflammation. The relative contribution of defects in mitochondrial maintenance and retrograde transport of STING to an ARF1-dependent type I interferonopathy disease-state remains to be determined. Although each process may be sufficient to cause disease[14,15,49], it is possible that these two aberrant activities drive the mutant-associated phenotype synergistically, i.e. stimulation of the cGAS-STING pathway via mtDNA is further exacerbated by defective signal termination. Improper termination of cGAS-STING triggering in the presence of ARF1 R99C, e.g. after exogenous triggers like viral infection, might accelerate and aggravate disease progression. Future studies dissecting the different functions of ARF1 in the regulation of the IFN response to infectious diseases, and the molecular mechanism by which ARF1 maintains mitochondrial integrity, are warranted.

Our data add to the emerging evidence that impaired control of cGAS-STING signal activation and termination is central to the pathogenesis of a number of type I interferonopathies[4]. The first described example of a Mendelian disease associated with chronic type I IFN signalling was Aicardi–Goutières syndrome (AGS). Mutations in TREX1, the RNase H2 complex, the deoxynucleoside triphosphate triphosphohydrolase SAMHD1 and the U7 small nuclear RNP complex all result in IFN induction by self DNA[4]. Recently, we described pathogenic mutations in mitochondrial ATPase family AAA domain-containing protein 3 A (ATAD3A) to result in a leakage of mtDNA into the cytosol[49]. Furthermore, defects in STING trafficking along the ER-Golgi axis have been reported to underlie various type I interferonopathy diseases. For example, activating mutations in STING cause SAVI[50–52], where in vitro studies revealed that mutations in the oligomerisation interface of STING (e.g. N154S, V155M and V147L, G207E, R281Q, R284G and R284S) result in spontaneous cGAMP-independent accumulation of STING at the Golgi, and aberrant induction of type I IFN signalling. Related to this, dysregulation of STING retrieval from the Golgi, due to heterozygous missense mutations in a component of COPI, coatomer protein subunit alpha (COPA), lead to enhanced type I IFN signalling[14,15,53]. Here, STING has been suggested to be recruited to COPA-containing COPI vesicles via SURF4[15,17], although other players may also be involved in regulating and triggering retrograde transport of STING. Besides recycling as a mechanism of signal termination, active STING is captured for subsequent autophagy-dependent degradation[11,13,54]. Recently, it was shown that ARF1 may also have a role in autophagy. However, our data show that cGAS-STING activation by the ARF1 R99C mutation is not mediated by autophagy[55]. Future studies are needed to genetically dissect the relative contribution of autophagy-mediated STING turnover and ARF1-mediated STING recycling to overall cGAS/STING signal termination.

Our data indicate a general defect in retrograde trafficking in the presence of ARF1 R99C (Fig. 5g, h). Besides STING recycling, retrograde trafficking retrieves key factors required for ER export[56,57], and is central to ER and Golgi homeostasis[58]. It is thus likely that non-IFN mediated mechanisms also contribute to the disease phenotype

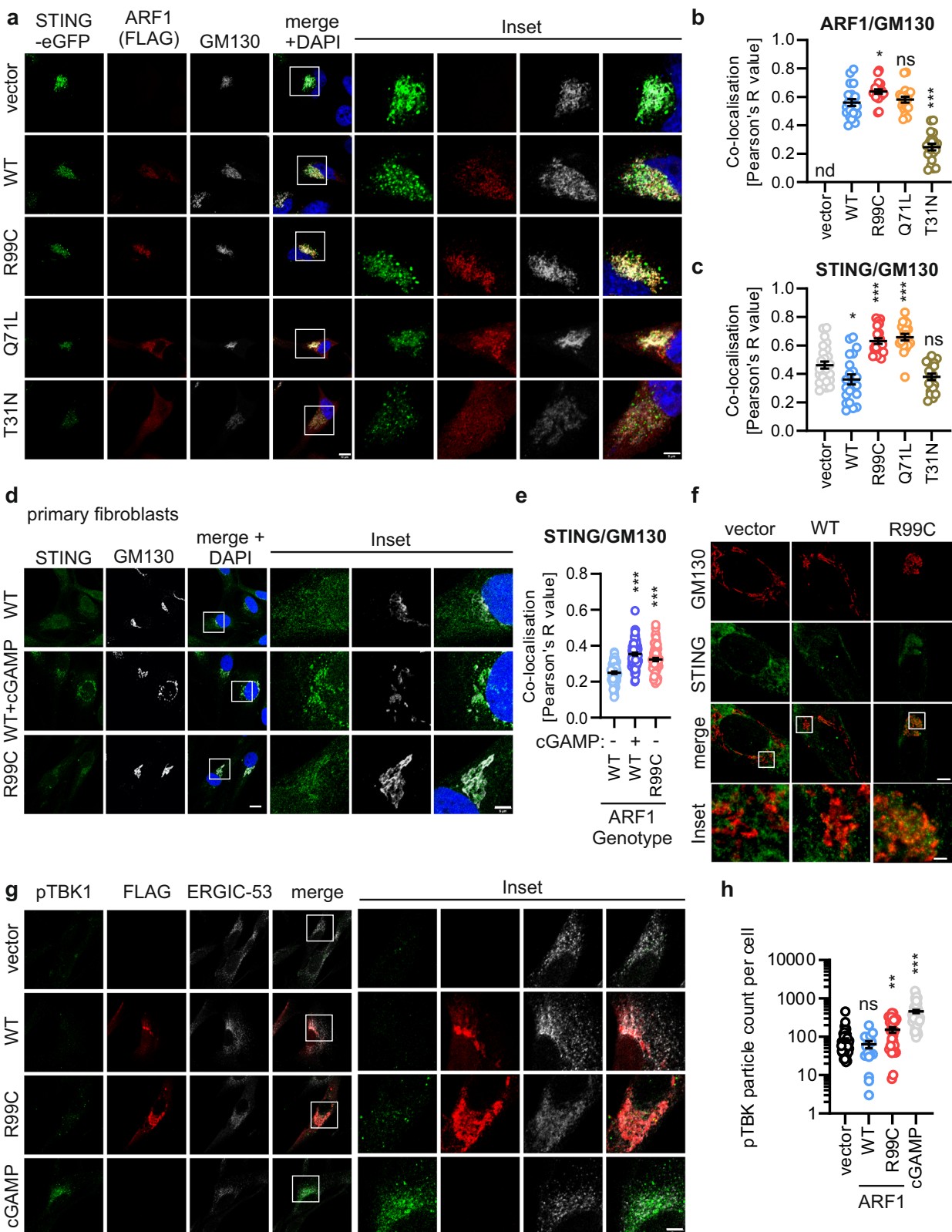

reported here, and that clinical disease caused by mutations in ARF1 may be variably accompanied by elevated type I IFN signalling. That being said, the recording of elevated levels of ISGs in patient blood, and the skin lesions present in three of the four cases that we ascertained (Fig. 1), are highly characteristic of other Mendelian type I interferonopathies[4,5]. Some functions of ARF1 are redundant and may be compensated by other ARF proteins[56,59,60]. However, any such

compensation is clearly insufficient in the case of the heterozygous mutations at R99 of ARF1. We speculate that complete loss of ARF1 activity would not be viable. Of note, similar to other interferonopathies with a defect in STING trafficking, such as the COPA syndrome, one defective allele is sufficient for disease[14]. Thus, ARF1 R99C and other disease-causing mutations in the STING trafficking pathway may act in a dominant negative fashion in regards to DNA

**Fig. 6 | STING accumulates at the ERGIC in the presence of ARF1 R99C.**
**a** Exemplary confocal laser scanning microscopy images of HeLa cells transiently expressing STING-eGFP (green) and FLAG-tagged ARF1 mutants (red). Cells were stained 24 h post transfection with anti-FLAG and anti-GM130 (grey). Nuclei, DAPI (blue). Scale bar, 10 μm or 5 μm (inset). **b** Quantification of the co-localisation of ARF1 and GM130 and **c** STING and GM130 from the images shown in (**a**), as Pearson's correlation coefficient. Lines represent mean of $n = 26$ (vector), $n = 21$ (WT, R99C), $n = 19$ (Q71L), $n = 20$ (T31N) ± SEM (individual cells). **d** Exemplary confocal laser scanning microscopy images of primary fibroblasts from healthy donors or patient. cGAMP (10 μg/ml, 3 h). Cells were stained with anti-STING (green) and anti-GM130 (grey). Nuclei, DAPI (blue). Scale bar, 10 μm or 5 μm (inset). **e** Quantification of the co-localisation of STING and GM130 from (**d**) as Pearson's correlation coefficient. Lines represent mean of $n = 71$ (WT), $n = 70$ (WT + cGAMP), $n = 75$

(R99C) ± SEM (individual cells). **f** Exemplary STED super-resolution microscopy images primary human lung fibroblasts (NHLF) transduced with lentiviruses expressing ARF1 constructs or empty vector. Cells were stained with anti-STING (green), anti-FLAG (not shown) and anti-GM130 (red) 48 h post transduction. Scale bar, 5 μm or 1 μm (inset merge, bottom panel). The experiment was repeated two times independently to similar results. **g** pTBK1 (green) in NHLF cells transduced with lentiviruses expressing indicated FLAG-tagged ARF1 constructs. Cells were stained with anti-FLAG (red) and anti-ERGIC-53 (white). cGAMP (10 μg/ml, 3 h). Scale bar, 10 μm or 5 μm (inset). **h** Quantification of the area of pTBK1 in (**g**). Lines represent mean of $n = 50$ (vector), $n = 16$ (WT), $n = 27$ (R99C), $n = 61$ (cGAMP) ± SEM (individual cells). See also Supplementary Fig. 6. Statistical analysis was performed using two-tailed Student's t test with Welch's correction. $*p < 0.05$; $**p < 0.01$; $***p < 0.001$. Exact $P$ values and Source data are provided in the Source data file.

sensing, while the heterozygous presence of WT ARF1 may be sufficient to maintain organismal viability.

In summary, through an exploration of the mechanistic basis of a human type I interferonopathy, our data establish key roles of the small GTPase ARF1 in both ensuring mitochondrial integrity and in Golgi-ER retrograde STING trafficking to prevent aberrant cGAS-STING pathway activity. These data highlight the importance of recycling mechanisms in innate immune homeostasis, and provide further evidence towards involvement of Golgi-associated ARF GTPases in mitochondrial integrity. As such, our results may inform a targeted treatment approach to the ARF-dependent type I interferonopathy state.

## Methods
### Ethics statement
Clinical information and samples were obtained with informed consent and consent to publish this research. Consent to participate in research and consent to publish the findings were obtained either from the patients or, in the case of underage patients, from their respective parent(s) or guardian(s). The study was approved by the Comite de Protection des Personnes (ID-RCB/EUDRACT: 2014-A01017-40) in France, and the Leeds (East) Research Ethics Committee (REC reference: 10/H1307/2 IRAS project ID: 62971) in the UK. ARF1 patient cells were handled with approval of the Ethics Committee at Ulm University (Approval 530/21). Sex and/or gender was not considered in the study design and is of no relevance top the study. Information on each individual patient is provided in the Supplementary Information file.

### Cell culture
HEK293T (ATCC, CRL-3216), HEK293T ATG5 KO, Hela (ATCC, CCL-2), 293-Dual-hSTING-R232 (Invivogen, 293d-r232), A549-Dual (Invivogen, a549d-nfis) cell lines, normal human lung fibroblast primary cells (Lonza, CC-2512) and normal human dermal fibroblasts primary cells (Thermo Fisher, C0045C; Innoprot, P10856; Promocell, C-12300) were cultivated in Dulbecco's modified Eagle medium (DMEM) supplemented with 10% (v/v) foetal bovine serum (FBS), 100 U/ml penicillin, 100 μg/ml streptomycin, and 2 mM L-glutamine. THP-1-Dual (Invivogen, thpd-nfis) and THP-1-Dual KO-cGAS (Invivogen, thpd-kocgas) cells were cultivated in RPMI 1640 medium supplemented with 10% (v/v) foetal bovine serum (FBS), 100 U/ml penicillin, 100 μg/ml streptomycin, and 2 mM L-glutamine. U2OS cells (ATCC, HTB-96) were maintained in McCoy medium (Invitrogen, 16600082) supplemented with 10% (v/v) foetal bovine serum. All cells were incubated at 37 °C in a 5% $CO_2$, 90% humidity atmosphere. Patient cells are from AGS640 were isolated by skin punch biopsy and cultivated as the other dermal fibroblast cells.

### Expression constructs and cloning
A construct coding for human ARF1 (pCMV6-hARF1-myc-FLAG) was purchased from Origene (PS100012, kindly provided by Michaela Gack, Florida). Mutations in R99C, R99H, R99A, R99E, R99K, Q71L and T31N were introduced by Q5 site-specific mutagenesis (see primers in

Table 1). Constructs coding for cGAS 3×-FLAG and STING-FLAG were kindly provided by Jae U. Jung (University of Southern California). STING-FLAG mutations R238A/Y240A and S366A were introduced by Q5 site-specific mutagenesis (see primers in Table 1). pEGFP-VSVG was a gift from Jennifer Lippincott-Schwartz (Addgene plasmid # 11912[61]). ECFP-ELP1-25 was a gift from Michael Davidson (Addgene plasmid # 55341). The open reading frame (ORF) of KDELR (ELP1) was subcloned into the pEGFP-VSVG vector using Gibson assembly (New England Biolabs, E5520S). The insert was amplified by PCR (see primers in Table 1) and the vector linearized with EcoRI and ApaI restriction enzymes. The ORF of TagRFP (from pCR3-TagRFP) was subcloned into

## Table 1 | Primers used for cloning

| Name | Sequence 5'–3' |
| --- | --- |
| ARF1-Q71Lfwd | TTG GAC AAG ATC CGG CCC CTG TGG |
| ARF1-Q71Lrev | GCC ACC CAC GTC CCA CAC AGT |
| ARF1-T31N-fwd | AAT ACG ATC CTC TAC AAG CTT AAG CTG GGT GAG A |
| ARF1-T31N-rev | CTT CCC TGC AGC ATC AGG GCC |
| ARF1 R99Cfwd | GAG TGT GTG AAC GAG GCC CGT GAG GAG C |
| ARF1 R99Crev | TCT GTC ATT GCT GTC CAC CAC GAA GAT C |
| SDM-ARF1-R99K/E/A R | TTG CTG TCC ACC ACG AAG |
| SDM-ARF1-R99A-F | TGA CAG AGA GGC TGT GAA CGA GGC CC |
| SDM-ARF1-R99E-F | TGA CAG AGA GGA GGT GAA CGA GGC C |
| SDM-ARF1-R99K-F | TGA CAG AGA GAA GGT GAA CGA GGC C |
| STING-R238A/Y240A-fwd | CAT CAA GGA TGC GGT TGC CAG CAA CAG CAT C |
| STING-R238A/Y240A-rev | CCA GCA CGG TCA GCG GTC TG |
| STING-S366A-fwd | GCT CCT CAT CGC TGG AAT GGA AAA GCC C |
| STING-S366A-rev | TCA GGC TCT TGG GAC ATC |
| VSVG(tsO45)-KDELR-fwd | GAC TTG GAA ACA GAA TTC TGA TGG CCA TGA ACA TTT CC G |
| VSVG(tsO45)-KDELR-rev | CTC ACC ATT GGA TCC CGG GCC CCT GCT GGC AAA CTG AGC TTC T |
| ARF1-TagRFP-fwd | GTC CAA TCA GCT CCG GAA CCA GAA GGC GGT GTC TAA GGG CGA AG |
| ARF1-TaqRFP-rev | CAG CTA TGA CCG CGG CCG GCC GT T TTT AAT TAA GTT TGT GCC CC |
| pBOB-ARF1-fwd | CCT CCA TAG AAG ACA CCG ACT CTA GAG CCA CCA TGG GGA ACA TCT TCG CC |
| pBOB-ARF1-rev | CTA TGA CCG CGG CCG GCC GTT TAA ACC TTA TCG TCG TCA TCC |
| pBOB-IRES-Puro-fwd | ACG GCC GGC CGC GGT CAT AGG CGG CCG CTC TAG CCC AAT TCC |
| pBOB-IRES-Puro-rev | GCT CCA TGT TTT TCC AGG TTT TCA GGC ACC GGG CTT GCG |
| EGFP-STING-fwd | ATT ACT CGA GAT GCC CCA CTC CAG |
| EGFP-STING-rev | GAA TTC TCA AGA GAA ATC CGT GCG GA |

the pCMV6-hARF1-myc-FLAG vector and the pCMV6-hARF1-R99C-myc-FLAG vector using Gibson assembly. The insert was amplified by PCR (see primers in Table 1) and the vectors were linearized with MluI and PmeI restriction enzymes. To insert the ORFs of ARF1 WT or ARF1 R99C (from pCMV6-ARF1) in a lentiviral backbone, both ORFs together with IRES-Puro (from pIRES-TRIM2-FLAG) were subcloned into the pBoB-hCas9-IRES-Bla vector using Gibson assembly. The inserts were amplified by PCR (see primers in Table 1) and the vector was linearized with XbaI and PmeI restriction enzymes. Constructs coding for human ARFGAP1 (pCMV3-ARFGAP1-HA, HG15006-CY) and VCP (pCMV3-VCP-HA, HG17232-CY) were purchased from Sino Biologicals. pGAPDH_PROM_01_Renilla SP Luciferase, pISRE-FLuc, pIFNb-luc, and TK-RL plasmids were described previously[62–64]. Human STING ORF was amplified from pMSCV-hygro-STING plasmid (Addgene #102598) by PCR (see primers in Table 1) and inserted into linearized pEGFP-C3 (Clontech) using XhoI and EcoRI enzymes. pCMV6-ARFGEF1-TurboGFP was purchased from Cliniscences (RG222817).

## Transfection of mammalian cells
DNA of expression vectors was transiently transfected using either the TransIT-LT1 Transfection Reagent (Mirus, MIR 2300) or Poly-ethylenimine (PEI, 1 mg/ml in $H_2O$) according to the manufacturers recommendations or as described previously[65].

## Transduction of mammalian cells
pMDLg, RSV-Rev, and pMD.G together with the generated pBOB constructs were used to rescue 3rd generation lentiviruses[62,65]. Cells were incubated with 3rd generation lentiviral particles for 16 h. Subsequently, the cells were washed three times with DMEM and incubated for further 48 h.

## Whole-cell lysates
Whole-cell lysates were prepared by harvesting cells in Phosphate-Buffered Saline (PBS, Gibco, 14190144). If not mentioned otherwise, the cell pellet ($500 \times g$, 4 °C, 5 min) was lysed in transmembrane lysis buffer (150 mM NaCl, 50 mM 4-(2-hydroxyethyl)-1-piper-azineethanesulfonic acid (HEPES) pH 7.4, 1% Triton X-100, 5 mM ethylenediaminetetraacetic acid (EDTA)) by vortexing at maximum speed for 30 s[65]. Cell debris was removed by centrifugation ($20,000 \times g$, 4 °C, 20 min), and the protein concentration of the supernatants was quantified using a BCA assay (Pierce Rapid Gold BCA Protein Assay Kit, Thermo Fisher Scientific, A53225). The lysates were then stored until analysis at −20 °C.

## SDS-PAGE and immunoblotting
SDS-PAGE and immunoblotting was performed using standard techniques[65]. In brief, whole-cell lysates were mixed with 6x Protein Sample Loading Buffer (at a final dilution of 1x) supplemented with 15% β-mercaptoethanol, heated to 95 °C for 5 min, separated on NuPAGE 4–12% Bis-Tris Gels (Invitrogen, NP0321BOX) for 90 min at 90 V and blotted onto Immobilon-FL PVDF membranes (Merck Millipore, 05317-10EA). The transfer was performed at a constant voltage of 30 V for 30 min. After the transfer, the membrane was blocked in 1% Casein in PBS. Proteins were stained with primary antibodies mouse anti-FLAG M2 (1:5000, Sigma-Aldrich, F1804), sheep anti-STING (1:1000, Bio-Techne, MAB7169), rabbit anti-pTBK1 (1:1000, Cell Signaling, 5483), rabbit anti-TBK1 (1:1000, Cell Signaling, 3504), rabbit anti-IRF3 (1:1000, Cell Signaling, 4302), rabbit anti-HA (1:1000, Cell Signaling, 3724), rabbit anti-ARF1 (1:300, Proteintech, 20226-1-AP), rabbit anti-cGAS (1:2000, Proteintech, 26416-1-AP), rabbit anti-RFP (1:1000, Abcam, ab28664), mouse anti-turboGFP (1:1000, Origene, TA150041S), rabbit anti-TFAM (1:1000, Proteintech, 22586-1-AP), mouse anti-Lamin B1 (1:10,000, Proteintech, 66095-1-Ig), rat anti-GAPDH (1:1000, BioLegend, 607902), rabbit anti-MFN1 (1:1000, Cell Signaling, 14739), mouse anti-RHOT1 (1:1000, Abnova, H00055288-M01), mouse anti-Drp1

(1:1000, Cell Signaling, 8570), rabbit anti-pDrp1 (1:1000, Cell Signaling, 3455), rabbit anti-COPB (1:1000, Abcam, Ab2899) and subsequently Infrared Dye labelled secondary antibodies (LI-COR, IRDye 680RD Goat anti-Mouse IgG Secondary Antibody, 926-68070; IRDye 680RD Goat anti-Rabbit IgG Secondary Antibody, 926-68071; IRDye 800CW Goat anti-Mouse IgG Secondary Antibody, 926-32210; IRDye 800CW Goat anti-Rabbit IgG Secondary Antibody, 926-32211; IRDye 800CW Goat anti-Rat IgG Secondary Antibody, 926-32219), diluted in 0.05% Casein in PBS. Band intensities were quantified using Image Studio lite (LI-COR).

## Luciferase reporter assays
HEK293T cells were either transiently transfected with luciferase reporter constructs, or reporter cell lines (A549-Dual, 293-Dual hSTING-R232, THP-1-Dual or THP1-Dual KO-cGAS) were used. 32 h post-transfection or 72 h post transduction, cells were lysed in passive lysis buffer (Promega, E1941) and luciferase activities of the firefly luciferase (FFLuc), renilla luciferase, lucia luciferase or SEAP activity were determined[64]. For HEK293T cells ISRE-firefly luciferase activities normalised to renilla activity were measured via DualGlo Luciferase Assay System (Promega, E2980). For the reporter cell lines ISRE-lucia luciferase activity (IFNb-lucia luciferase for 293-Dual hSTING-R232 cells) was measured 1 s after injecting 20 mM coelenterazine (PFK Biotech, 102173) and NF-κB-SEAP activity (ISRE-SEAP for 293-Dual hSTING-R232 cells) via Alkaline Phosphatase Blue Microwell Substrate (Sigma-Aldrich, AB0100). Both were normalized to cell viability determined by the CellTiter-Glo Luminescent Cell Viability Assay (Promega, G7570). Luciferase and cell viability measurements were performed using an Orion II microplate Luminometer and the Simplicity software (Berthold), SEAP activity was measured at 650 nm by using a Vmax kinetic microplate reader (Molecular Devices) and the SoftMax Pro 7.0.3 software.

## Generation of U2OS cells stably expressing STING and mtDNA depletion
The pMSCV-hygro plasmid carrying WT STING1 cDNA (Addgene plasmid #102598) or empty vector were used in combination with packaging vector pCL-Ampho (Novus) and envelope vector pCMV-VSV-G (Addgene plasmid #8454) to produce retroviral vectors as described for lentiviral vectors. 100,000 U2OS cells were transduced with 0.5 mL retroviral vectors, 8 µg/mL polybrene (Millipore, TR-1003-G) and 10 mM HEPES (Invitrogen, H3375) in 12-well plates, and medium replaced 24 h later. Two days after transduction, transduced cells were selected and maintained in culture with 200 µg/mL hygromycin B. STING expression was verified by western blotting. mtDNA depletion in U2OS and U2OS-STING cells was induced by 100 µM 2′,3′ dideoxycytidine (ddC, Sigma-Aldrich, D5782) treatment in medium supplemented with 50 µg/mL uridine (Sigma-Aldrich, U3750) and 1 mM sodium pyruvate (Gibco, 11360070) for seven to fourteen days before use. To control for mtDNA depletion, total DNA from 500,000 cells was extracted using the DNeasy Blood and Tissue Kit (Qiagen, 69506), following the manufacturer's instructions. DNA concentrations were determined by photometry (Nanodrop) and 15 ng and 7.5 ng DNA were used to perform qPCR for the mitochondrial gene MT-COXII and the nuclear gene GAPDH (see primers in Table 2). Quantitative PCR (qPCR) was performed using Power SYBR Green (Invitrogen, A25742). Ratios of ΔΔCt for MT-COXII over GAPDH for the different DNA concentrations were averaged and the fold change to untreated (UT) is shown in figures. U2OS cells were stimulated for 4 h with 2 µg/mL HT-DNA (Sigma-Aldrich, D6898) complexed with Lipofectamine 2000 (Invitrogen, 11668019), according to the manufacturer's instructions, or lipofectamine alone.

## Generation of HEK293T ATG5 KO cells
HEK293T ATG5 KO cells were generated by genomic knock out using CRISPR Cas9. To this end, 3rd generation lentiviral vectors were

**Table 2 | Primers used for SYBR Green qPCR**

| Type | Primers | Sequence |
|------|---------|----------|
| SYBR Green | MT-COXII_F | CGTCTGAACTATCCTGCCCG |
| SYBR Green | MT-COXII_R | TGGTAAGGGAGGGATCGTTG |
| SYBR Green | GAPDH_F | ATGCTGCATTCGCCCTCTTA |
| SYBR Green | GAPDH_R | GCGCCCAATACGACCAAATC |
| SYBR Green | KCNJ10 fwd | GCGCAAAAGCCTCCTCATT |
| SYBR Green | KCNJ10 rev | CCTTCCTTGGTTTGGTGGG |
| SYBR Green | MT-Dloop fwd | CATAAAGCCTAAATAGCCCACACG |
| SYBR Green | MT-Dloop rev | CCGTGAGTGGTTAATAGGGTGATA |

**Table 3 | Primers used for TaqMan qPCR**

| Type | Primers | Assay ID |
|------|---------|----------|
| TaqMan | HPRT1 | Hs03929096_g1 |
| TaqMan | IFI27 | Hs01086370_m1 |
| TaqMan | RSAD2 | Hs01057264_m1 |
| TaqMan | OAS1 | Hs00973637_m1 |
| TaqMan | MX1 | Hs00895608_m1 |
| TaqMan | IFNB1 | Hs01077958_s1 |

generated as described before[65] using pSicoR-CRISPR-PuroR CRISPR/Cas9[66] constructs harbouring an ATG5 targeting sgRNA or a non-targeting (NT) sgRNA. (NT: ACGGAGGCTAAGCGTCGCAA, ATG5: AACTTGTTTCACGCTATATC)[67] as the transfer plasmid. HEK293T cells were transduced with the lentiviral vectors and 3 days post-transduction separated into individual cells using limited dilution. The individual cells were grown into clonal cell lines and screened for ATG5 KO using Western blot analysis (anti-ATG5 antibody, Cell Signaling Technology, #2630). Clones with a conformed knock out were expanded and stocks were conserved by cryo preservation.

## Overexpression of WT and mutant ARF1 in U2OS cells

U2OS WT and U2OS-STING cells plated in 12-well plates were transiently transfected with ARF1 WT, ARF1 R99C or empty vector. Cells were collected 24 h later for RNA and protein analysis. Total RNA was extracted using the RNAqueous-Micro Kit (Ambion, AM1931), and reverse transcription performed with the High-Capacity cDNA Reverse Transcription Kit (Applied Biosystems, 4368814). Levels of cDNA were quantified by RT-qPCR using TaqMan Gene Expression Assay (Applied Biosystems). Differences in cDNA inputs were corrected by normalization to HPRT1 cDNA levels. Relative quantitation of target cDNA was determined by the formula $2^{-\Delta\Delta CT}$ (see Taqman probes in Table 3). For whole-cell lysate analysis, proteins were extracted from U2OS cells using RIPA lysis buffer with 1% protease inhibitor and 1% phosphatase inhibitor. Bolt LDS Sample Buffer (4X, Novex Life Technologies) and Bolt Sample Reducing agent (10X, Novex Life Technologies) were added to protein lysates, samples resolved on 4–12% Bis-Tris Plus NuPAGE gels (Invitrogen, B0007) and then transferred to nitrocellulose membrane for 7 min at 20 V using the iBlot 2 Dry Blotting System (Invitrogen). To analyse protein phosphorylation status, membranes were blocked in LI-COR buffer, and primary phospho-antibodies (rabbit anti-pIRF3, 1:1000, Cell Signaling, 4947; rabbit anti-pSTING, 1:1000, Cell Signaling, 19781) incubated for 48 h in blocking solution. For cofilin immunoblot, membranes were blocked with 5% non-fat milk in TBS, and primary antibodies (rabbit anti-Cofilin, 1:1000, Cell Signaling, 5175) incubated overnight at 4 °C in 1.5% Bovine Serum Albumin in TBS buffer supplemented with 0.1% Tween. After stripping, membranes were reblotted with anti-STING antibodies (mouse anti-STING, 1:1000, R&D Systems, MAB7169; rabbit anti-IRF3, 1:1000, Cell Signalling, 4302) in 2.5% non-fat milk in TBS buffer supplemented with 0.1% Tween. After washing, membranes were incubated with appropriate anti-mouse or anti-rabbit secondary antibodies for 45 min at room temperature (LI-COR). Signal was detected using the OdysseyCLx System (LI-COR) and Image Studio software. Comparative signal analyses were performed using Fiji (ImageJ).

## Assessing mitochondrial DNA release into the cytosol

HEK293T WT or HEK293T ATG5 KO cells were transfected with ARF1 WT, ARF1 R99C or empty vector. Alternatively, HEK293T WT cells were transfected with ARF1 WT, ARF1 R99C or empty together with VCP or empty vector. 24 h later, cells were treated with 10 µM of ABT-737 (SYNkinase, 1001) and 10 µM

Quinoline-Val-Asp-Difluorophenoxymethylketone (Q-VD-OPH, Cayman Chemical, 15260) as a positive control. In the case of primary human dermal fibroblasts, cells from four healthy donors or from patient AGS460 were used. On the next day, the cells were harvested and isolation and quantification of DNA from cytosolic, mitochondrial and nuclear fractions was performed as described previously[68] (basic protocol 2). Briefly, half of the cells were lysed in SDS lysis buffer (20 mM Tris, pH 8, 1% (v/v) SDS, protease inhibitors) to obtain WCLs for normalisation, whereas the other half was used for fractionation. Cytosolic, mitochondrial and nuclear extracts were isolated by subsequently incubating the cells with saponin lysis buffer (1x PBS, pH 7.4, 0.05% saponin, protease inhibitors), NP-40 lysis buffer (50 mM Tris, pH 7.5, 150 mM NaCl, 1 mM EDTA, 1% (v/v) NP-40, 10% (v/v) glycerol, protease inhibitors) and SDS lysis buffer (20 mM Tris, pH 8, 1% (v/v) SDS, protease inhibitors), respectively. Purity of the fractions was determined by immunoblotting for GAPDH (cytosolic extract), TFAM (mitochondrial extract), and Lamin B1 (nuclear extract). DNA extraction of the fractions and WCLs was performed using phenol-chloroform. DNA concentrations were determined by photometry (Nanodrop) and equal amounts of DNA were used to perform qPCR for mitochondrial DNA (MT-Dloop) and nuclear DNA (KCNJ10) (see primers in Table 2). qPCR was performed using PowerUP SYBR Green (Applied Biosystems, A25742) and the relative cytosolic mtDNA was calculated using the $\Delta\Delta CT$ method.

## Expression of WT and mutant ARF1 in NHLF cells

NHLF cells were transduced with lentiviral particles coding for ARF1 WT, ARF1 R99C or empty vector. For qPCR analysis, total RNA was extracted 72 h post transduction using the Quick-RNA Microprep Kit (Zymo research, R1051) according to the manufacturer's instructions. Reverse transcription and qRT–PCR were performed in one step using the SuperScript III Platinum Kit (Thermo Fisher Scientific, 11732088) on a StepOnePlus Real-Time PCR System (Applied Biosystems) according to the manufacturer's instructions. TaqMan probes for each individual gene were acquired as premixed TaqMan Gene Expression Assays (Thermo Fisher Scientific) and added to the reaction (See probes in Table 3). Expression levels for each target gene were calculated, e.g. for OAS1 expression levels by normalizing to GAPDH cDNA levels using the $\Delta\Delta CT$ method.

## Dimerization assay

HEK293T cells were transfected with ARF1 WT (pCMV6-ARF1-myc-FLAG or pCMV6-ARF1-Tag-RFP, or with pCMV6-ARF1-myc-FLAG and pCMV6-ARF1-Tag-RFP) or ARF1 R99C (pCMV6-ARF1-R99C-myc-FLAG or pCMV6-ARF1-R99C-Tag-RFP, or with pCMV6-ARF1-R99C-myc-FLAG and pCMV6-ARF1-R99C-Tag-RFP). 24 h post transfection, WCLs were prepared and input samples were saved for western blotting. The WCLs were incubated with anti-FLAG M2 magnetic beads (Sigma-Aldrich, M8823) for 4 h at 4 °C on a rotating shaker. Subsequently, the beads were washed five times with transmembrane lysis buffer and incubated with 1x Protein Sample Loading Buffer supplemented with 15% β-mercaptoethanol. After heating to 95 °C for 10 min the samples were applied to SDS-PAGE and immunoblotting.

## In vitro GTPase assays

HEK293T cells were transfected with pCMV6-ARF1-myc-FLAG, pCMV6-ARF1-R99C-myc-FLAG, pCMV6-ARF1-Q71L-myc-FLAG or pCMV6-ARF1-T31N-myc-FLAG. 24 h later, WCLs were prepared in GTPase lysis buffer (150 mM NaCl, 50 mM HEPES pH 7.4, 1% Triton X-100, 5 mM MgCl$_2$, 5 mM EDTA) and incubated with anti-FLAG M2 magnetic beads (Sigma-Aldrich, M8823) for 4 h at 4 °C on a rotating shaker. Subsequently, the beads were washed three times with washing buffer I (500 mM NaCl, 50 mM HEPES pH 7.4, 1% Triton X-100, 5 mM MgCl$_2$, 5 mM EDTA) and twice with washing buffer II (100 mM NaCl, 50 mM HEPES pH 7.4, 1% Triton X-100, 5 mM MgCl$_2$, 5 mM EDTA). Next, the beads were incubated with GTPase-Glo-GEF assay buffer (Promega, V7681) and the GTPase reaction was performed according to the manufacturer's recommendations (GTPase-Glo assay, Promega, V7681). In short, the beads were incubated with 2x GTP solution for 16 h followed by the addition of the GTPase-Glo reagent and the detection reagent. The resulting luminescence was measured using an Orion II microplate Luminometer and the Simplicity software (Berthold).

## Stable isotope labelling of amino acids in cell culture (SILAC)

To analyse interaction partners of ARF1 WT and ARF1 R99C, stable isotope labelling of amino acids in cell culture (SILAC)-based quantitative mass spectrometry (MS) was performed. HEK293T cells were cultivated in SILAC medium light (DMEM for SILAC (Thermo Fisher Scientific, 88364) supplemented with 10% (v/v) dialysed FBS, 100 U/ml penicillin, 100 µg/ml streptomycin, 2 mM L-glutamine, 200 mg/ml proline (Thermo Fisher Scientific, 88211), 84 mg/ml L-arginine (Thermo Fisher Scientific, 88427), 146 mg/ml L-lysin (Thermo Fisher Scientific, 89987)) or SILAC medium heavy (DMEM for SILAC supplemented with 10% (v/v) dialysed FBS, 100 U/ml penicillin, 100 µg/ml streptomycin, 2 mM L-glutamine, 200 mg/ml proline (Thermo Fisher Scientific, 88211), 87.2 mg/ml 13C15N-labelled L-arginine (Carl Roth, 2063.1), 152.8 mg/ml 13C15N-labelled L-lysin (Carl Roth, 2085.1)) for five passages to completely incorporate the labelled amino acids. Complete incorporation was controlled by analysing a heavy sample prior to immunoprecipitation as described below. Incorporation levels reached 85% and no proline conversion was observed. The cells were either transfected with ARF1 WT (heavy-labelled cells) or ARF1 R99C (light-labelled cells). 24 h later, WCLs were prepared using transmembrane lysis buffer (150 mM NaCl, 50 mM 4-(2-hydroxyethyl)-1-piperazineethanesulfonic acid (HEPES) pH 7.4, 1% Triton X-100, 5 mM ethylenediaminetetraacetic acid (EDTA)) from $1 \times 10^7$ cells and incubated with anti-FLAG M2 magnetic beads (Sigma-Aldrich, M8823) for 4 h at 4 °C on a rotating shaker. Subsequently, the beads were washed five times with transmembrane lysis buffer and incubated with 1x Protein Sample Loading Buffer supplemented with 15% β-mercaptoethanol. After heating to 95 °C for 10 min the samples were applied to SDS-PAGE and immunoblotting or MS.

## Mass spectrometry (MS) and data analysis

SILAC labelled samples were combined in a 1-to-1 manner and proteins were separated using standard 12.5% SDS-Page followed by colloidal Coomassie staining and subsequent sample preparation as described earlier[69]. Samples were measured using an LTQ Orbitrap Elite system (Thermo Fisher Scientific) online coupled to an U3000 RSLCnano (Thermo Fisher Scientific) employing an Acclaim PepMap™ analytical column (ID: 75 µm × 500 mm, 2 µm, 100 Å, Thermo Fisher Scientific) in combination with a C18 µ-precolumn (0.3 mm inner diameter (ID) × 5 mm; PepMap, Dionex LC Packings, Thermo Fisher Scientific). Samples were preconcentrated and washed with 0.1% TFA for 5 min at a flow rate of 30 µl/min. Subsequent separation was carried out employing a flow rate of 250 nl/min using a binary solvent gradient consisting of solvent A (0.1% FA) and solvent B (86% ACN, 0.1% FA). The main column was initially equilibrated in 5% B. The percentage of B was raised from 5 to 15% in 10 min, followed by an increase from 15 to 40% B in 20 min.

The column was washed with an increase to 95% B in 5 min and holding at 95% B for 5 min. Finally, the column was re-equilibrated with 15% B for 20 min.

The mass spectrometer was equipped with a nanoelectrospray ion source and distal coated SilicaTips (FS360-20-10-D, New Objective). The instrument was externally calibrated using standard compounds (LTQ Velos ESI Positive Ion Calibration Solution, Pierce, Thermo Scientific) The system was operated using the following parameters: spray voltage, 1.5 kV; capillary temperature, 250 °C; S-Lens RF Level, 68.9%. XCalibur 2.2 SP1.48 (Thermo Fisher Scientific) was used for data-dependent MS/MS analyses. Full scans ranging from m/z 370 to 1700 were acquired in the Orbitrap at a resolution of 30,000 (at m/z 400) with Automatic gain control (AGC) enabled and set to $10^6$ ions and a maximum fill time of 500 ms. For fragmentation in the linear ion trap the AGC was set to 10,000 ions and a maximum fill time of 100 ms. For MS/MS fragmentation of the top 20 most intense ions, a normalized collision energy of 35% with an activation q of 0.25 and an activation time of 30 ms was used. The resulting fragments were analysed using the LTQ part at rapid scan speeds.

Database search was performed using MaxQuant Ver. 1.6.3.4 (www.maxquant.org)[70]. For peptide identification and quantitation, MS/MS spectra were correlated with the UniProt human reference proteome set (www.uniprot.org, Version on March 16th 2021), supplemented with the ARF1 sequences, employing the build-in Andromeda search engine[71]. The respective SILAC modifications and carbamidomethylated cysteine were considered as fixed modification along with oxidation (M), and acetylated protein N-termini as variable modifications. False discovery rates were set on both, peptide and protein level, to 0.01. Subsequent data analysis was performed employing MS Excel and GraphPad Prism 9. For the final analysis the mean of two replicates was considered and proteins with a mean log2 ratio of >1 or <−1 were considered as regulated.

## Co-immunoprecipitation

HEK293T cells were transfected with ARF1 WT, ARF1 R99C or vector control. 48 h post transfection, WCLs were prepared and input samples were saved for western blotting. The WCLs were incubated with anti-FLAG M2 magnetic beads (Sigma-Aldrich, M8823) for 4 h at 4 °C on a rotating shaker. Subsequently, the beads were washed five times with transmembrane lysis buffer (300 mM salt) and incubated with 1x Protein Sample Loading Buffer supplemented with 15% β-mercaptoethanol. After heating to 95 °C for 10 min the samples were analysed by SDS-PAGE and immunoblotting

## Go-term analysis

The top 30 genes less associated with ARF1 R99C compared to ARF1 WT according to the SILAC experiment were submitted to PantherDB[72,73]. Analysis Type: PANTHER Overrepresentation Test (Released 20220202). GO Ontology database DOI: 10.5281/zenodo.6399963, Released 2022-03-22. Reference List:Homo sapiens (all genes in database).

## Immunofluorescence

Cells were seeded on coverslips in 24-well plates and treated as indicated. Next, the samples were washed with PBS and fixed in 4% paraformaldehyde solution (PFA) for 20 min at RT, permeabilized and blocked with PBS containing 0.5% Triton X-100 and 5 FCS for 1 h at RT. Afterwards, the cells were washed with PBS and incubated for 2 h at 4 °C with primary antibody (mouse anti-FLAG M2, 1:400, Sigma-Aldrich, F1804; rabbit anti-GM130, 1:400, Cell Signaling, 12480; rabbit anti-ERGIC-53, 1:400, Proteintech, 13364-1-AP; sheep anti-TGN46, 1:400, Bio-Rad, AHP500GT; rabbit anti-pTBK1, 1:100, Cell Signaling, 5483; mouse anti-STING, 1:100, Novus Biologicals, AF6516; rabbit anti-STING, 1:200, Proteintech, 19851-1-AP) diluted in PBS with 1% FCS. After

washing with PBS/0.1% Tween 20, the samples were incubated with the secondary antibody donkey anti-mouse IgG (H + L) Alexa Fluor Plus 568 (A10037), donkey anti-mouse IgG (H + L) Alexa Fluor Plus 488 (A32766), donkey anti-rabbit IgG (H + L) Alexa Fluor Plus 647 (A32795), donkey anti-sheep IgG (H + L) Alexa Fluor Plus 647 (A-21448 (1:400, Thermo Fisher Scientific) or with primary antibody-secondary antibody conjugates (rabbit anti-ERGIC-53, 1.25 μg/ml; rabbit anti-GM130, 0.585 μg/ml; rabbit anti pTBK1, 1.42 μg/ml; rabbit anti-STING, 3.5 μg/ml; mouse anti-FLAG M2, 0.35 μg/ml; conjugated to equal amounts (μg/ml) of Zenon Alexa Fluor 647 rabbit IgG labelling reagent (Z25308), Zenon Alexa Fluor 568 rabbit IgG labelling reagent (Z25306), Zenon Alexa Fluor 488 rabbit IgG labelling reagent (Z25002) or Zenon Pacific Blue mouse IgG$_{2a}$ labelling reagent (Z25313, Thermo Fisher Scientific) and 500 ng/ml DAPI (Invitrogen, D1306) for 2 h at 4 °C in the dark. Next, the samples were washed with PBS/0.1% Tween 20 and water and the coverslips were mounted onto microscopy slides. Images were acquired using a Zeiss LSM 710 confocal laser scanning microscope with ZEN 2010 imaging software (Zeiss). Images were analysed with ImageJ (Fiji). The number of ERGIC-53 positive particles and the particle size was analysed using a custom ImageJ macro (Fiji). Colocalization was determined with the Huygens Professional 19.04 software. In short, Pearson coefficients were calculated with the "Huygens Colocalization Analyzer" using the Costes method[74] and applying individual thresholds.

### Live cell imaging of mitochondria and mitochondrial analysis

HeLa cells were seeded in 35 mm μ-Dishes (Ibidi, 81156) and transfected with TagRFP-labelled ARF1 WT, ARF1 R99C or vector control. 24 h later, all cells were treated with 1 μM Mitotracker (Thermo Fisher Scientific, M22426) and 1 μg/mL Hoechst 33342 (Thermo Fisher Scientific, 62249) for 30 min at 37 °C. Sequentially the medium was removed and exchanged by fresh medium without phenol red. Images were then acquired using a Zeiss LSM 710 confocal laser scanning microscope with ZEN 2010 imaging software (Zeiss). Images analysis was performed with ImageJ (Fiji). For analysis of the mitochondrial footprint, the background subtractor tool (MOSAIC group) and the MiNA plugin (StuartLab) were used. First, the background subtractor (length = 20) was used, then the despeckle ("Despeckle", "slice") command and finally single cells were examined using the MiNA Analyse morphology plugin (threshold=moments)[75]. The values obtained for the mitochondrial footprint (area/volume consumed by mitochondrial signal) were then used for visualization. Mitochondrial characteristics (perimeter, circularity) and network connectivity (branch length, branches per cell) were analysed as reported elsewhere[76]. In short, perimeter and circularity of the mitochondria were analysed by applying an auto threshold ("Triangle dark no-reset"), followed by the Convert to mask ("Convert to Mask", "method=Triangle background=Dark calculate black"), despeckle ("Despeckle", "slice") and remove outliers ("Remove Outliers…", "radius=2 threshold=50 which=Bright slice") commands and single cells were examined via the analyse particles ("Analyze Particles…", "clear summarize") command. Total branch length and branches per cell were analysed by applying the Skeletonize Command ("Skeletonize", "slice"). Single cells were isolated and examined using the Analyze skeleton command ("Analyze Skeleton (2D/3D)", "prune=none").

### VSVG transport assay

Retrograde transport of VSVG-ts045-KDELR was performed as described previously[45]. In brief, HeLa cells were transfected with pEGFP-VSVG-ts045-KDELR and ARF1 WT, ARF1 R99C or empty vector. The cells were incubated at 37 °C for 24 h and then directly fixed or incubated at 32 °C for 2 h to accumulate the fusion protein at the Golgi complex. Cells were then either fixed or shifted to 40 °C for 1 h to allow one round of retrograde transport from the Golgi to the ER and then fixed.

### STED sample preparation

Normal human lung fibroblasts (NHLF) were transduced with lentiviral particles coding for ARF1 WT, ARF1 R99C or empty vector. 48 h later, the samples were washed with PBS and fixed in 4% paraformaldehyde solution (PFA) for 20 min at RT. Next, the cells were permeabilized and unspecific binding was blocked by incubation with blocking solution (3% (w/v) BSA and 0.3% (v/v) Triton X-100 in PBS) for 2 h at RT. The samples were incubated overnight at 4 °C with 1 μg/ml of the primary antibodies rabbit anti-GM130 (Cell Signaling, 12480) and mouse anti-STING (Novus Biologicals, AF6516) dissolved in diluted blocking solution (0.3% (w/v) BSA and 0.03% (v/v) Triton X-100 in PBS). After three washing steps with PBS, the samples were incubated with 1 μg/ml secondary goat anti-mouse antibody conjugated with Atto647N (Sigma-Aldrich, 50185), 1 μg/ml goat anti-rabbit antibody conjugated with Atto594 (Sigma-Aldrich, 77671) and anti-rat antibody conjugated with Alexa Fluor Plus 405 (Thermo Fisher Scientific, A48268, transfection control) dissolved in diluted blocking solution (0.3% (w/v) BSA and 0.03% (v/v) Triton X-100 in PBS) for 1 h at RT. Unbound antibodies were removed in three washing steps with PBS. For imaging, samples were kept in 97% 2,2′-thiodiethanol (TDE, Sigma Aldrich, 166782) solution in PBS, pH 7.5.

### STED imaging

Images were captured with a home-built dual-colour 3D-STED microscope with home-built software[77]. Typically, an average power of ~1 μW for each excitation beam (568 nm and 633 nm, respectively) and ~1.5 mW for each depletion beam (710 nm and 750 nm, respectively) was used. STED images were captured at a pixel size of 20 nm and a dwell time of 300 μs with a typical peak photon number of ~150 counts. Images were analysed by ImageJ (Fiji). For better visualization, a Gaussian blur of σ = 1 was applied in each channel.

### EM preparation

Sample preparation was performed according to a standardized protocol[78]. HEK293T cells were cultivated on UV-sterilized 160 μm thin carbon-coated sapphire disks (Engineering Office M.) and transfected with ARF1 WT, ARF1 R99C or empty vector. 24 h later, the samples were then cryo-fixed at a pressure of 230 MPa within 30 ms using a high-pressure freezer (HPF Compact 101). The samples were freeze substituted in a medium of acetone with 0.1% uranyl acetate (UA), 0.2% osmium tetroxide (OsO4), and 5% double distilled water for improved visibility of the membranes[79]. Overnight (17 h), samples were gradually warmed in an EM AFS2 (Leica Microsystems GmbH) freeze substitution device from −90 to 0 °C. They were then left at 0 °C for 1 h and washed 3 times with acetone for 30 min each at room temperature and embedded in EPON resin (Sigma-Aldrich, 45345). For embedding, samples were incubated successively for one hour each in 33%, 50% and 67% EPON resin in acetone, then overnight in 100% EPON and polymerized for 48 h at 60 °C. By plunging the solidified specimens in liquid nitrogen, the EPON block breaks in the region of the embedded sapphire discs leaving the cells on the surface ready to be sectioned with an ultramicrotome (Ultracut UC7, Leica Microsystems GmbH).

### EM imaging

For TEM imaging, 70 nm thin sections were mounted on carbon-coated Formvar films on copper grids (Plano GmbH) and imaged with a JEM-1400 TEM operating at 120 kV acceleration voltage equipped with a CCD camera (Veleta, Olympus Life Science). For STEM tomography, 800 nm thick sections were put on glow-discharged copper grids with parallel bars (Plano GmbH), pre-treated with 10% (w/v) poly-L-lysine (Sigma-Aldrich), followed by a second coating with poly-L-lysine to attach 25 nm colloidal gold particles (AURION Immuno Gold Reagents) on both sides of the cross-sections. A series of tilted images of the sections from an angle of −72° to +72° with an increment of 1.5° were recorded using a Jeol FEM 2100 F field-emission TEM equipped with a

Jeol STEM bright-field detector (Jeol Ltd) and, EM-Menu 4.0 STEM tomography software (TVIPS) at a resolution of 1024 px × 1024 px, an illumination time of 20 s, and an acceleration voltage of 200 kV. Alignment of the images with the gold particles as fiducial markers as well as 3D-reconstruction of the tilt series was done using the IMOD 4.9 software[80,81].

## EM stereology

According to the Delesse principle, stating that the volume density of an organelle or component in a tissue can be estimated by measuring the area fraction of the intersections of the component within a random section of the tissue[82,83], volume fractions of luminal structures and small vesicles were determined by counting grid points on pre-defined classes within square test fields of 16 μm2 size using the recursive grid option implemented in the open-source software JMicroVision image analysis system. Since it is not possible to evaluate the entire cell, the precondition for stereological evaluation was that the section through the cell contained centrioles. The test fields were chosen so that the centrioles were located in their centre. This ensured that only similar areas in the cell were evaluated, as the distribution of organelles may differ depending on the cell area.

## Structural analysis

A model of the ARF1 crystal structure (2J59) retrieved from the Protein Data Bank (PDB) and visualised in UCSF Chimera 1.15. Only Chain A was displayed and R99C was highlighted by displaying the atom model.

## Purification of ARF1, ARF1 R99C and ARF1 Q71L

For bacterial protein expression, soluble human ARF1 wild-type and mutants lacking its N-terminal 17 amino acids and human ARFGAP1 domain (1–136) were cloned into modified pET16 vector with N-terminal His$_6$-MBP-SUMO tag. *E. coli* BL21 Rosetta cells were grown at 37 °C in lysogenic broth medium until the culture reached an OD$_{600}$ of 0.4–0.5 and protein production was induced at 18 °C with 0.4 mM isopropyl-β-thiogalactopyranoside (Sigma-Aldrich, I6758) for 16 h. Harvested *E. coli* cells were resuspended in lysis buffer (20 mM HEPES pH 7.5, 400 mM NaCl, 30 mM imidazole, 10% glycerol and 1 mM β-mercaptoethanol) and lysed by sonication. Recombinant cell debris was removed by centrifugation and recombinant hARF1 proteins were purified over nickel-nitriloacetic acid (Ni-NTA, Qiagen, 30250) affinity chromatography and the His$_6$-MBP-SUMO tag was subsequently removed by addition of SENP2 protease at 4 °C, followed by overnight dialysis against 20 mM HEPES pH 7.5, 250 mM NaCl and 2 mM β-mercaptoethanol. The proteins were further purified and separated from His$_6$-MBP-SUMO tag and protease by a HiLoad 16/600 Superdex 75 size exclusion chromatography column (Cytiva) in 20 mM HEPES pH 7.5, 250 mM NaCl, 1 mM TCEP. Purified hARF1 were pooled, aliquoted and flash frozen in liquid nitrogen before stored at −80 °C.

## Thermal shift assay

The thermal stability of different hARF1 proteins in presence and absence of GTP was analysed by fluorescence thermal shift assays. 75 μM protein were incubated in 25 mM HEPES pH 7.5, 100 mM NaCl, 5 mM MgCl$_2$, 1 mM TCEP with or without 5 mM GTP. The fluorescence signal was detected after addition of SYPRO orange (final concentration 5x, Invitrogen, S6651) using gradient from 15 °C to 95 °C with 0.5 °C/30 s and one scan each 0.5 °C in a real time thermal cycler (QuantStudio 3, Thermo Fisher Scientific). The deflection point of the curve and first derivative was calculated by Prism 9 (GraphPad).

## Autophagy reporter assay

The autophagosome levels of HEK293T cells stably expressing LC3B-GFP were assessed by flow cytometry. HEK293T cells were transfected with Tag-RFP ARF1 WT, Tag-RFP ARF1 R99C or empty vector using PEI. 6 h after transfection the medium was changed to reduce effects of

transfection reagents on the cells. For samples that were treated with Bafilomycin A1, Bafilomycin A1 (Santa Cruz Biotechnology, sc-201550) at a concentration of 625 μM was added to the medium. 24 h after treatment, the samples were detached and transferred to 96-well V-bottom plates. Treatment with 0.05% saponin in PBS and two subsequent washes with PBS were used to remove cytosolic LC3B-GFP. Fluorescence intensity of membrane-bound LC3B-GFP was measured using a Beckman-Coulter CytoFLEX with attached high-throughput sampler and set above 1000 to allow for detection of shifts in autophagosome levels in both directions (more or less autophagosomes). Intact single cells were gated using SSC-A/FSC-A and FSC-A/FSC-H, respectively. Raw fluorescence-activated cell sorting (FACS) data were analysed using FlowJo 10. Median fluorescence intensity shifts of all samples were calculated by subtracting the LC3B-GFP-MFI of vector-treated samples from the ARF1 WT and ARF1 R99C samples.

## Quantification and statistical analysis

Statistical analyses were performed using GraphPad Prism 9. *P*-values were determined using a two-tailed Student's *t* test with Welch's correction or one-way ANOVA for multiple comparisons (Mann–Whitney test). Statistics on qPCRs over multiple values (Fig. 3f and Supplementary Fig. 3c–f) were performed using two-way ANOVA. Unless otherwise stated, data are shown as the mean of at least three biological replicates ± SEM. Significant differences are indicated as: $*p < 0.05$; $**p < 0.01$; $***p < 0.001$. Not significant differences are not indicated. Specific statistical parameters are specified in the figure legends.

## Reporting summary

Further information on research design is available in the Nature Portfolio Reporting Summary linked to this article.

## Data availability

The SILAC experiments were deposited to the MassIVE database under accession code MSV000089711. Source data are provided in the Supplementary Information and Source data files. Source data are provided with this paper.

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

## Acknowledgements

We thank Jana-Romana Fischer, Birgit Ott, Regina Burger, Daniela Krnavek, Kerstin Regensburger, Martha Meyer and Nicola Schrott for excellent technical assistance. This study was supported by DFG (German Research Foundation) grants SP1600/4-1 (to K.M.J.S.), SPP1923 (to K.M.J.S. and F.K.), CRC1279 (to K.M.J.S., F.K., J.M., C.R., S.W. and P.W.), as well as the BMBF to F.K. (Restrict SARS-CoV-2) and K.M.J.S. (IMMUNOMOD 01KI2014) and the DFG Emmy Noether Programme 458004906 to CCOM. Y.J.C. and K.M.J.S. are additionally supported by a DFG/Agence Nationale de la Recherche (ANR, France) joint grant (504830917). Y.J.C. acknowledges the European Research Council (786142 E-T1IFNs), a UK Medical Research Council Human Genetics Unit core grant (MC_UU_00035/11), a state subsidy from the Agence Nationale de la Recherche (France) under the 'Investissements d'avenir' programme bearing the reference ANR-10-IAHU-01, and MSDA-VENIR (Devo-Decode Project). Y.J.C. sincerely thanks Elizabeth Sztul and Paul Randazzo for the generation of early preliminary data not included in the manuscript, and Marine Depp and Carolina Uggenti for providing the eGFP-STING plasmid. Y.J.C. also thanks Paolo Piccolo for expert clinical phenotyping, Ignazia Prigione for experimental support, Gillian Rice for deriving interferon scores, and Jean-Madeleine de Sainte Agathe for clinical advice. This study makes use of data generated by the DECIPHER community, with a full list of contributing centres available [https://www.deciphergenomics.org/about/stats] and via email from contact@deciphergenomics.org. S.V. acknowledges funding from the Italian Ministry of University and Research (PRIN grant n.20175XHBPN). Funding for the DECIPHER project was provided by the Wellcome Trust. A.L. acknowledges Inserm International Research Project (Inserm IRP) programme. M.H., L.K., S.K. and V.H. are part of the international graduate school in molecular medicine, Ulm (IGradU).

## Author contributions

Y.J.C., A.L., M.H. and K.M.J.S. conceived the project, and designed and interpreted experiments. M.H. performed most of the experiments. A.L., V.H., L.K., S.K., V.M., T.M., M.P.R. and B.D.B. performed additional experiments. U.R., T.B., C.R. and P.W. contributed the electron microscopy studies. F.W. and J.M. provided super-resolution imaging. S.W. and R.R. analysed and performed the mass spectrometry analysis for the SILAC experiment. V.M. and C.C.d.O.M. performed and planned the in vitro experiments. C.C.d.O.M., J.M. and F.K. supervised experiments and helped interpret data. S.V., M.G., R.P., S.A.L., M.G.H., G.H., K.M.W., J.S. and J.L. contributed patient data. Y.J.C., A.L., M.H. and K.M.J.S. wrote the manuscript.

## Funding

## Competing interests

The authors declare no competing interests.

## Additional information

[1]Institute of Molecular Virology, Ulm University Medical Center, 89081 Ulm, Germany. [2]Université Paris Cité, Imagine Institute, Laboratory of Neurogenetics and Neuroinflammation, INSERM UMR1163, F-75015 Paris, France. [3]Central Facility for Electron Microscopy, Ulm University, 89081 Ulm, Germany. [4]Institute of Virology, Technical University of Munich, 81675 Munich, Germany. [5]Institute of Biophysics, Ulm University, 89081 Ulm, Germany. [6]Core Unit Mass Spectrometry and Proteomics, Ulm University, 89081 Ulm, Germany. [7]UOC Reumatologia e Malattie Autoinfiammatorie, IRCCS Istituto Giannina Gaslini, Genoa, Italy. [8]Università degli Studi di Genova, Genoa, Italy. [9]Children's Health Ireland, Crumlin, Dublin, Eire. [10]University College Dublin, Dublin, Eire. [11]Department of Medical Genetics, St. Olav's Hospital, Trondheim, Norway. [12]Department of Medical Genetics, Haukeland University Hospital, 5021 Bergen, Norway. [13]Division of Genomic Medicine, Department of Pediatrics, University of California, Davis in Sacramento, CA, USA. [14]Rady Children's Institute for Genomic Medicine, San Diego, CA, USA. [15]Division of Pediatric and Adolescent Dermatology, Rady Children's Hospital San Diego, San Diego, CA, USA. [16]Department of Dermatology, University of California San Diego School of Medicine, La Jolla, USA. [17]MRC Human Genetics Unit, Institute of Genetics and Cancer, University of Edinburgh, Edinburgh, UK. ✉e-mail: yanickcrow@mac.com; Konstantin.Sparrer@uni-ulm.de

