## [Peer Review File · Nature Communications]

REVIEWER COMMENTS

Reviewer #1 (Remarks to the Author):

Overall, the authors have done a substantial amount of work to experimentally demonstrate that the R99C mutation in ARF1 disrupts mitochondrial fusion and impairs retrograde transport of STING, together resulting in elevated type I interferon signaling in vitro and in patients. The authors have effectively addressed reviewer comments and concerns and have included new data that strengthens the evidence supporting their conclusions

We note that experimental evidence for a type I interferonopathy has only been provided for R99C, with ambiguous clinical findings for the patient described to have the R99H mutation and without any experimental results provided for this variant. Two individuals heterozygous for the W78* ARF1 mutation have previously been published in a report that includes pictures of their face and hands which do not demonstrate evidence of chilblain lupus (PMID 34353862). Indeed, while neurologic impairment appears to be shared across individuals with an identified ARF1 mutation (this manuscript, PMID 34353862, and 28868155 referenced by the authors), the elevated type I interferon signature has only been demonstrated for the R99C mutation. In light of this, it may be more accurate to introduce a caveat for the R99H mutation indicating that it remains unclear whether it causes elevations in type I interferon signaling. This could be clarified in the discussion to indicate that findings refer only to R99C rather than R99 more generally (example lines 362,363, 427, etc).

Indeed, it is interesting to speculate that various ARF1 mutations may cause different clinical phenotypes through impairing ARF1 in different ways. This may explain why prior studies have linked inhibition or depletion of ARF1 to impaired anterograde STING transport, consistent with a dampened immune response, as opposed to the findings described in this work. In the context of an elevated type I interferon signature, as would be expected, it is the retrograde transport of STING that is impaired. Given this heterogeneity, however, it may be prudent to clearly indicate that clinical disease (developmental delay) caused by mutations in ARF1 may or may not be accompanied by an elevated type I interferon signature, particularly as mutations other than R99C were not investigated by the authors.

Minor point: in line 266 the authors list COPS8 as a member of the COPI vesicle rather than COPZ1 which is likely a typo as COPS8 is a COP9 signalosome member while COPZ1 codes for the coat complex subunit ζ .

Reviewer #2 (Remarks to the Author):

In this article, Hirschenberger et al have performed (among many other investigations) affinity purification -MS experiments to assess the interactome of ARF1 WT and an ARF1-mutant (R99C)

The IP experiments were performed in a rather standard way by transfecting HEK293 cells with FLAG tagged constructs followed by anti-FLAG IP.

To enable accurate quantitation the authors performed the experiments in Heavy / Light SILAC labelled cells. Proteins eluted from IP beads were mixed, fractionated by SDS-PAGE, in-gel digested and analysed by LC-MS/MS.

MS data analysis was performed using the MaxQuant software with standard parameters.

***General evaluation of proteomics experiments/data

Overall the experimental workflow seems in itself sound. The results - at least as described in the main text of the article and figures - appear to show the loss of key interactions by the mutant Arf1 vs the WT and this would corroborate the main hypothesis of the manuscript.

Despite this initially positive impression, upon closer inspection some problems appear. First, there are significant deficiencies in how the proteomics experiments are described and even much less clarity in the description of the data processing and analysis. Closer examination of Supplementary Table 2, which is supposed to report the SILAC MS data left me puzzled and raised a number of questions on both the experimental design, the data analysis that was applied and the conclusions that were drawn from the results. Without necessarily impacting the remaining findings of the manuscript, what I observed makes me question if the proteomics data presented can lead to the conclusions that the authors have drawn from them.

***Detailed comments

Experiments are described on page 11-12 (lines 258-265). The text gives the impression that a direct comparison of cells transfected with WT ARF1 vs WT was performed but in fact very little detailed information is provided.

However, Supplem. Table 2 shows 4 series of columns, 2 labelled RggC (= R99C presumably) and 2 labelled ARF1. The columns containing the H/L ratios, which is normally the main measure generated by SILAC, are mostly empty. This suggests that to me that in fact 4 experiments were done in which either the WT or the mutant were SILAC compared each to an empty control. Interestingly, the authors do not give any explicit description of how the comparisons were constructed so this remains rather opaque.

This experimental design, although feasible in principle, would not be an ideal choice for the biological question. SILAC works best when the conditions of interest are directly compared. Even more so for AP-MS experiments in which a negative empty control is present, as this tends to result in many missing ratios (proteins absent in one condition) so that the negative control does not work well as a common reference sample. For the purpose of the comparison WT vs R99C this transform the SILAC experiments essentially into separate label-free experiments.

A more serious problem appears when one looks at the values for Arf1. Quantitation of ARF1 is not straightforward because of the presence of the WT and mutant sequences which largely overlap, but also because of the high sequence homology with ARF3 (and the fact that MaxQuant only uses razor/unique peptides for quantitation). Even when these caveats are taken into account, it appears that the protein was detected with high intensity in the R99C IPs, while the signal in the WT replicates (?) was hundred of folds lower than in the mutants. This is also obvious from a separate worksheet the authors provide, listing ARF1 peptides and their intensities.

This is a major flaw of the experiment because the data in the two conditions are not really comparable with such a huge difference in amount of bait protein. In essence, if my interpretation is correct, either transfection or immunoprecipitation of the WT protein did not work as it should have done.

By the way, the authors appear to use the total absolute Intensity values for the WT ARF1 and mut Arf1 (2 replicates each) to calculate ratios ("calculations" worksheet). This is the summed intensity for the H and L channels in a given SILAC comparison and in my opinion should not be used in the way it has been used (although it does not change dramatically the overall situation).

Incidentally, the authors state in the text that COPI proteins showed " a ~3400-fold lower association with ARF1 R99C" (line 266). In my experience it is very rare that such extreme ratios can effectively be measured even by SILAC - which is assumed to be one of the most accurate proteomics techniques. I suppose that this surprising value is the result of "ratio of ratios" calculations done with heavily skewed (100's of fold) values, an insufficient number of replicates and some technically failed samples (also, it remains not clear how the values presented in Fig. 4f are derived from the data in Supplem. Table 2).

When I look at COPI proteins in Supplem. Table 2, it is true that they show similar values in all the R99C and WT IPs, while there is >200x less ARF1 recovered in the WT samples, which taken literally may lead to the conclusion that much less COPI is associated with the mutant. However it is also possible that the COPI levels observed are just unspecific background that remains roughly constant in all IPs, irrespective of ARF1 presence or not.

Again, in my opinion it is not possible to compare IPs in which the bait is present at dramatically different levels, as this may lead to strongly biased conclusions. So in my opinion the data presented is in part flawed and does not allow to draw the conclusions that the authors report.

Note : the issues with the proteomics dataset do not imply that the rest of the conclusions and data of the manuscript are wrong. This only suggests that the proteomics dataset is not exploitable to answer the specific question on the interactions of WT Arf1 vs R99C Arf1.

Other points :

- again, the description on how the MaxQuant output was processed is sparse or absent (as discussed above). This is not acceptable. Reviewers and readers should not have to undertake forensic investigations in supplementary tables to assess the robustness of the findings.

- the criteria used for filtering are also left largely obscure , apart from the fact the the significance B parameter was used. This is a rather outdated method.

- the standard for proteomics experiments today is to perform 3 or more replicates and validate data by regular statistical tests (T-test, Welch's test). This experiments do not seem to satisfy these criteria.

- no mention is made of deposition of the raw MS data and full MaxQuant output into a public repository (proteomexchange.org). This is also a standard requirement today, though it should have been checked by the journal in the initial phase of the submission...? Having other MaxQuant output files (summary.txt, parameters.txt) available would have been useful for an evaluation.

Minor points

Methods section : additional information / clarification is needed on :

- mention whether completeness of SILAC labelling was assessed and how; state if Arg to Pro conversion was detectable
- specify origin of SILAC medium and SILAC AA's
- specify the composition of buffer used for obtaining "WCL"
- LC-MS : specify at least the length of the gradients used, the flow rate and the diameter of the column.
- specify the company providing (or a reference on) the software "Origin Pro" used in data analysis

Reviewer #3 (Remarks to the Author):

cGAS-STING pathway in interferon production and signalling is understood in considerable details. STING, an ER protein is trafficked to the Golgi, where it is involved in events leading to phosphorylation of IRF3 and the activation of NF-KB to mediate interferon production and export. The role of Arf1 and COPI and retrograde traffic from ERGIC/Golgi is well documented. The potentially new aspect in this paper is the effect of mutations in Arf1 on mitochondrial organisation. How does this happen? The authors are encouraged to address this novel aspect.

**Point-by-point response to the reviewers' comments on manuscript no. NCOMMS-23-16659-T
"ARF1 prevents aberrant type I IFN induction by regulating STING activation and recycling"**

Reviewer #1 (Remarks to the Author):

Overall, the authors have done a substantial amount of work to experimentally demonstrate that the R99C mutation in ARF1 disrupts mitochondrial fusion and impairs retrograde transport of STING, together resulting in elevated type I interferon signaling in vitro and in patients. The authors have effectively addressed reviewer comments and concerns and have included new data that strengthens the evidence supporting their conclusions

We thank the reviewer for her/his positive assessment of our data.

We note that experimental evidence for a type I interferonopathy has only been provided for R99C, with ambiguous clinical findings for the patient described to have the R99H mutation and without any experimental results provided for this variant. Two individuals heterozygous for the W78* ARF1 mutation have previously been published in a report that includes pictures of their face and hands which do not demonstrate evidence of chilblain lupus (PMID 34353862). Indeed, while neurologic impairment appears to be shared across individuals with an identified ARF1 mutation (this manuscript, PMID 34353862, and 28868155 referenced by the authors), the elevated type I interferon signature has only been demonstrated for the R99C mutation. In light of this, it may be more accurate to introduce a caveat for the R99H mutation indicating that it remains unclear whether it causes elevations in type I interferon signaling. This could be clarified in the discussion to indicate that findings refer only to R99C rather than R99 more generally (example lines 362,363, 427, etc).

The interferon signature shown in Fig. 1j is exemplary from a patient with the R99H mutation. Patients with the R99C mutation also show an IFN signature (compare Supplementary Patient Data). We have now more clearly indicated the ARF1 mutation status of the patients in the manuscript.

To further clarify whether R99 is critical for the ability of ARF1 to prevent aberrant IFN signaling, we constructed and analyzed additional ARF1 mutants (R99H, R99A, R99E and R99K). Expression of all of them resulted in an upregulation of type I IFN signaling equivalent to the ARF1 R99C substitution (new Supplementary Fig. 2b, c). These data further support our conclusion that R99 is critical for ARF1-mediated suppression of type I IFN signaling.

Indeed, it is interesting to speculate that various ARF1 mutations may cause different clinical phenotypes through impairing ARF1 in different ways. This may explain why prior studies have linked inhibition or depletion of ARF1 to impaired anterograde STING transport, consistent with a dampened immune response, as opposed to the findings described in this work. In the context of an elevated type I interferon signature, as would be expected, it is the retrograde transport of STING that is impaired. Given this heterogeneity, however, it may be prudent to clearly indicate that clinical disease (developmental delay) caused by mutations in ARF1 may or may not be accompanied by an elevated type I interferon signature, particularly as mutations other than R99C were not investigated by the authors.

As outlined above, we have now examined a panel of substitutions at position 99, all of which are associated with an upregulation of interferon signalling. However, we completely agree with the reviewer that mutations at different positions might result in different phenotypic aspects, some of which may not be linked to elevated type I IFN signaling. We address this important point in the revised discussion section (line 438-440): "It is [thus] likely that non-IFN mediated mechanisms also contribute to the disease phenotype reported here, and that clinical disease caused by mutations in ARF1 may be variably accompanied by elevated type I IFN signalling."

Minor point: in line 266 the authors list COPS8 as a member of the COPI vesicle rather than COPZ1 which is likely a type as COPS8 is a COP9 signalosome member while COPZ1 codes for the coat complex subunit ζ .

The reviewer is correct, thank you. We have removed this, since the SILAC experiment was overhauled (see comments to Reviewer #2).

Reviewer #2 (Remarks to the Author):

In this article, Hirschenberger et al have performed (among many other investigations) affinity purification -MS experiments to assess the interactome of ARF1 WT and an ARF1-mutant (R99C) The IP experiments were performed in a rather standard way by transfecting HEK293 cells with FLAG tagged constructs followed by anti-FLAG IP.

To enable accurate quantitation the authors performed the experiments in Heavy / Light SILAC labelled cells. Proteins eluted from IP beads were mixed, fractionated by SDS-PAGE, in-gel digested and analysed by LC-MS/MS.

MS data analysis was performed using the MaxQuant software with standard parameters.

***General evaluation of proteomics experiments/data

Overall the experimental workflow seems in itself sound. The results - at least as described in the main text of the article and figures - appear to show the loss of key interactions by the mutant Arf1 vs the WT and this would corroborate the main hypothesis of the manuscript.

Despite this initially positive impression, upon closer inspection some problems appear. First, there are significance deficiencies in how the proteomics experiments are described and even much less clarity in the description of the data processing and analysis. Closer examination of Supplementary Table 2, which is supposed to report the SILAC MS data left me puzzled and raised a number of questions on both the experimental design, the data analysis that was applied and the conclusions that were drawn from the results. Without necessarily impacting the remaining findings of the manuscript, what I observed makes me question if the proteomics data presented can lead to the conclusions that the authors have drawn from them.

***Detailed comments

Experiments are described on page 11-12 (lines 258-265). The text gives the impression that a direct comparison of cells transfected with WT ARF1 vs WT was performed but in fact very little detailed information is provided.

However, Supplem. Table 2 shows 4 series of columns, 2 labelled RggC (= R99C presumably) and 2 labelled ARF1. The columns containing the H/L ratios, which is normally the main measure generated by SILAC, are mostly empty. This suggests that to me that in fact 4 experiments were done in which either the WT or the mutant were SILAC compared each to an empty control. Interestingly, the authors do not give any explicit description of how the comparisons were constructed so this remains rather opaque.

This experimental design, although feasible in principle, would not be an ideal choice for the biological question. SILAC works best when the conditions of interest are directly compared. Even more so for AP-MS experiments in which a negative empty control is present, as this tends to result in many missing ratios (proteins absent in one condition) so that the negative control does not work well as a common reference sample. For the purpose of the comparison WT vs R99C this transform the SILAC experiments essentially into separate label-free experiments.

We thank the new Reviewer #2 for pointing out this important issue. We have now reanalyzed the SILAC experiment starting from the raw data. Thus, we have removed a normalization on an empty vector control, which was included previously due to a communication error. The heavy (ARF1 WT)/light (ARF1 R99C) ratios are now shown as the means (new Fig. 4f). Only proteins which were detected in two independent experiments were considered for the analysis. No other normalizations were performed or cut-offs set. The sheet providing the raw data is now more clearly labelled (new Supplementary table 2), and the vector control experiments have been removed. Consequently, the PantherDB analysis in Fig. 4g has been reassessed as well (new Fig. 4g and new Supplementary table 3). The reanalyzed dataset still shows that COPI-vesicle components associated less to ARF1 R99C than WT, thus supporting the overall hypothesis of our manuscript. Our conclusions have been further strengthened by newly derived data relating to co-immunoprecipitation of ARF1 WT/R99C and endogenous COPB (new Fig. 4h).

A more serious problem appears when one looks at the values for Arf1. Quantitation of ARF1 is not straightforward because of the presence of the WT and mutant sequences which largely overlap, but also because of the high sequence homology with ARF3 (and the fact that MaxQuant only uses razor/unique peptides for quantitation). Even when these caveats are taken into account, it appears that the protein was detected with high intensity in the R99C IPs, while the signal in the WT replicates (?) was hundred of folds lower than in the mutants. This is also obvious from a separate worksheet the authors provide, listing ARF1 peptides and their intensities.

This is a major flaw of the experiment because the data in the two conditions are not really comparable with such a huge difference in amount of bait protein. In essence, if my interpretation is correct, either transfection or immunoprecipitation of the WT protein did not work as it should have done.

By the way, the authors appear to use the total absolute Intensity values for the WT ARF1 and mut Arf1 (2 replicates each) to calculate ratios ("calculations" worksheet). This is the summed intensity for the H and L channels in a given SILAC comparison and in my opinion should not be used in the way it has been used (although it does not change dramatically the overall situation).

Incidentally, the authors state in the text that COPI proteins showed "a ~3400-fold lower association with ARF1 R99C" (line 266). In my experience it is very rare that such extreme ratios can effectively be measured even by SILAC - which is assumed to be one of the most accurate proteomics techniques. I suppose that this surprising value is the result of "ratio of ratios" calculations done with heavily skewed (100's of fold) values, an insufficient number of replicates and some technically failed samples (also, it remains not clear how the values presented in Fig. 4f are derived from the data in Supplem. Table 2).

When I look at COPI proteins in Supplem. Table 2, it is true that they show similar values in all the R99C and WT IPs, while there is >200x less ARF1 recovered in the WT samples, which taken literally may lead to the conclusion that much less COPI is associated with the mutant. However it is also possible that the COPI levels observed are just unspecific background that remains roughly constant in all IPs, irrespective of ARF1 presence or not.

Again, in my opinion it is not possible to compare IPs in which the bait is present at dramatically different levels, as this may lead to strongly biased conclusions. So in my opinion the data presented is in part flawed and does not allow to draw the conclusions that the authors report.

Note : the issues with the proteomics dataset do not imply that the rest of the conclusions and data of the manuscript are wrong. This only suggests that the proteomics dataset is not exploitable to answer the specific question on the interactions of WT Arf1 vs R99C Arf1.

We apologize for our mistake in how we chose to display these data. The experiments labelled ctr were vector controls, i.e. endogenous ARF1 was detected, which is, as expected, much lower in abundance than ARF1 in the overexpression samples. For the new quantifications, we took all ARF1 peptides (i.e.

not only unique peptides) into account (see new Supplementary table 2), and show intensity vs mean log2 ratio WT/R99C as a new supplementary Figure 4f. As expected, the intensity of ARF1 is high (due to overexpression and enrichment by pulldown); however, the ratio between WT/R99, i.e. heavy and light-labelled ARF1, is at 1, indicating that the pulldown efficiency was similar. This is also supported by western blot staining of the pulldown samples (Supplementary Fig. 4d). After reanalysis, fold changes were recalculated based only on the H/L ratios, with the results section updated accordingly (line 274-282). ARF1 R99C now is 1.18, 1.32, 1.36 or 1.41-fold less associated with COPB1, COPA, COPG1 and COPB2, respectively.

Importantly, the SILAC data still show that ARF1 R99C associates less strongly with COPI vesicle components, thereby supporting our previous conclusions. Decreased binding of COPB to ARF1 R99C is now also further confirmed by co-immunoprecipitation of endogenous COPB from HEK293T cells and analysis by western blot (new Fig. 4h).

Other points :

- again, the description on how the MaxQuant output was processed is sparse or absent (as discussed above). This is not acceptable. Reviewers and readers should not have to undertake forensic investigations in supplementary tables to assess the robustness of the findings.

We have updated the Methods section to include more detail (line 657-712), and streamlined the Supplementary table containing the mass spectrometry data (new Supplementary table 2).

- the criteria used for filtering are also left largely obscure , apart from the fact the the significance B parameter was used. This is a rather outdated method.

Arbitrary criteria as correctly pointed out by the reviewer were removed, and ratios calculated in all cases where a heavy and light labeled protein was detected (new Supplementary Table 2). Only if the protein was detected in both independent repeats of the SILAC experiment was it considered for the final presentation (new Fig 4f and g).

- the standard for proteomics experiments today is to perform 3 or more replicates and validate data by regular statistical tests (T-test, Welch's test). This experiments do not seem to satisfy these criteria.

The experiment was performed twice independently, and our new analysis only includes proteins/ratios that were detected in both experiments (see new Supplementary Table 2, new Figs. 4f, 4g and new Supplementary Fig. 4f). The conclusions in the results section have been toned down accordingly (line 272-289). We have now, independently of the SILAC assays, verified altered binding of ARF1 R99C to endogenous COPI vesicles (COPB) using co-immunoprecipitation assays (new Fig. 4h).

- no mention is made of deposition of the raw MS data and full MaxQuant output into a public repository (proteomexchange.org). This is also a standard requirement today, though it should have been checked by the journal in the initial phase of the submission...? Having other MaxQuant output files (summary.txt, parameters.txt) available would have been useful for an evaluation.

We sincerely apologize for this oversight; the data were previously deposited on the MassIVE database (Accession # MSV000089711). Review-Account: MSV000089711_reviewer; Password: ARF1_review), but we failed to include this information in the manuscript. The record has now been altered to fit the new data analysis, and will be made publicly available upon acceptance of our manuscript. A data availability statement has been added to the manuscript (line 1093-1096).

Minor points

Methods section : additional information / clarification is needed on :

- mention whether completeness of SILAC labelling was assessed and how; state if Arg to Pro conversion was detectable

Both parameters were assessed. As assessed by analysis of the heavy-labelled input sample, the completeness of SILAC labeling was at 85% (see new Supplementary Table 2, sheet 2). No ARG to PRO conversion was detectable. This is now stated in the Methods section (line 657-712).

- specify origin of SILAC medium and SILAC AA's

DMEM for SILAC (Thermo Scientific, Cat No: 88364) supplemented with 10% (v/v) dialyzed FBS, 100 U/ml penicillin, 100 µg/ml streptomycin, 2 mM L-glutamine, 200 mg/ml proline (Thermo Scientific, Cat No: 88211), 84 mg/ml L-arginine (Thermo Scientific, Cat No: 88427), 146 mg/ml L-lysine (Thermo Scientific, Cat No: 89987) or SILAC medium heavy (DMEM for SILAC supplemented with 10% (v/v) dialyzed FBS, 100 U/ml penicillin, 100 µg/ml streptomycin, 2 mM L-glutamine, 200 mg/ml proline (Thermo Scientific, Cat No: 88211), 87.2 mg/ml 13C15N-labelled L-arginine (Carl Roth, Cat No: 2063.1), 152.8 mg/ml 13C15N-labelled L-lysine (Carl Roth, Cat No: 2085.1)). Please also see the updated methods section line 657-712.

- specify the composition of buffer used for obtaining "WCL"

We used Transmembrane lysis buffer (150 mM NaCl, 50 mM 4-(2-hydroxyethyl)-1-piperazineethanesulfonic acid (HEPES) pH 7.4, 1% Triton X-100, 5 mM ethylenediaminetetraacetic acid (EDTA). Please also see the updated Methods section line 657-712.

- LC-MS : specify at least the length of the gradients used, the flow rate and the diameter of the column.

Samples were measured using an LTQ Orbitrap Elite system (Thermo Fisher Scientific) online coupled to an U3000 RSLCnano (Thermo Fisher Scientific, Idstein, Germany) employing an Acclaim® PepMap™ analytical column (ID: 75 µm x 500 mm, 2 µm, 100 Å, Thermo Fisher Scientific, Bremen, Germany) in combination with a C18 µ-precolumn (0.3 mm inner diameter (ID) x 5 mm; PepMap, Dionex LC Packings, Thermo Fisher Scientific, Bremen, Germany). Samples were preconcentrated and washed with 0.1% TFA for 5 min at a flow rate of 30 µl/min. Subsequent separation was carried out employing a flow rate of 250 nl/min using a binary solvent gradient consisting of solvent A (0.1% FA,) and solvent B (86% ACN, 0.1 % FA). The main column was initially equilibrated in 5% B. The percentage of B was raised from 5% to 15% in 10 min, followed by an increase from 15% to 40% B in 20 min. The column was washed with an increase to 95% B in 5 min and holding at 95% B for 5 min. Finally, the column was re-equilibrated with 15 % B for 20 min. Please also see the updated Methods section line 657-712.

- specify the company providing (or a reference on) the software "Origin Pro" used in data analysis

The software Origin Pro was not used in the re-analysis and has now been removed. The database search was performed using MaxQuant Ver. 1.6.3.4 (www.maxquant.org). For peptide identification and quantitation, MS/MS spectra were correlated with the UniProt human reference proteome set (www.uniprot.org, Version on March 16th 2021), supplemented with the ARF1 sequences, employing the build-in Andromeda search engine. Please also see the updated Methods section line 657-712.

Reviewer #3 (Remarks to the Author):

cGAS-STING pathway in interferon production and signalling is understood in considerable details. STING, an ER protein is trafficked to the Golgi, where it is involved in events leading to phosphorylation of IRF3 and the activation of NF- κ B to mediate interferon production and export. The role of Arf1 and COPI and retrograde traffic from ERGIC/Golgi is well documented. The potentially new aspect in this paper is the effect of mutations in Arf1 on mitochondrial organisation. How does this happen? The authors are encouraged to address this novel aspect.

We agree with the reviewer that the impact of ARF1 on mitochondrial fusion is an interesting and novel aspect of our work. In addition to our existing data, we have now performed a further in-depth analysis of mitochondria dynamics in ARF1 WT or ARF1 R99C expressing cells. Specifically, we have analyzed mitochondrial footprint, perimeter and circularity, as well as the branch number and length, of the mitochondrial network (updated Supplementary Figs. 3j, 3k; new Figs. 3g, 3h). Our results show that the circularity and number of branches of the mitochondrial network is decreased in R99C ARF1 expressing cells, whereas the perimeter and branch length are increased. This indicates elongation of mitochondria. In agreement with our previous data showing that levels of the fusion factor MFN1 are significantly increased in the presence of ARF1 R99C (Fig. 3i), the new results strongly suggest that expression of ARF1 R99C results in excessive fusion of mitochondria, and their consequent destabilization.

REVIEWERS' COMMENTS

Reviewer #1 (Remarks to the Author):

The authors have effectively addressed our comments and concerns and have included new data that strengthens the evidence supporting their conclusions.

Reviewer #2 (Remarks to the Author):

The authors have addressed practically all the points I have raised in a satisfactory manner. The manuscript, supplementary materials and figures have been modified accordingly.

Reviewer #3 (Remarks to the Author):

I am still not satisfied with the explanation that Arf99C affects mitochondrial fusion. But the authors have done a lot of work to make the connection and I don't want to delay the publication of this paper. My gut feeling is that there are at least two parallel events in cells expressing Arf99C. One is the effect on retrograde transport, which is not surprising. The second on the mitochondrial physiology is confusing. Are ERGIC membranes the source of membranes for MFN clearance? Is there a general defect in autophagy or is this more specific for mitophagy? Would reducing MFN levels restore the detrimental effect of ARF99C on STING signalling?

I leave it to the authors to address this concern, but if this is not possible experimentally then please explain this potential caveat in the discussion.

**Point-by-point response to the reviewers' comments on manuscript no. NCOMMS-23-16659-T
"ARF1 prevents aberrant type I IFN induction by regulating STING activation and recycling"**

We thank all three reviewers for their support and comments.

Reviewer #1 (Remarks to the Author):

The authors have effectively addressed our comments and concerns and have included new data that strengthens the evidence supporting their conclusions.

Reviewer #2 (Remarks to the Author):

The authors have addressed practically all the points I have raised in a satisfactory manner. The manuscript, supplementary materials and figures have been modified accordingly.

Reviewer #3 (Remarks to the Author):

I am still not satisfied with the explanation that Arf99C affects mitochondrial fusion. But the authors have done a lot of work to make the connection and I don't want to delay the publication of this paper. My gut feeling is that there are at least two parallel events in cells expressing Arf99C. One is the effect on retrograde transport, which is not surprising. The second on the mitochondrial physiology is confusing. Are ERGIC membranes the source of membranes for MFN clearance? Is there a general defect in autophagy or is this more specific for mitophagy? Would reducing MFN levels restore the detrimental effect of ARF99C on STING signalling?

I leave it to the authors to address this concern, but if this is not possible experimentally then please explain this potential caveat in the discussion.

We thank the reviewer for supporting publication of our manuscript. We were also surprised by the impact of ARF1 on mitochondrial integrity. However, future studies are required to elucidate the detailed molecular mechanism of ARF1 in regulating mitochondrial fusion. We have now added a sentence to the discussion section (line 412-413).

The mechanism by which the status of ERGIC membranes impact MFN stability remains unclear. Data in yeast indicate that ARF1 may recruit the E3 ubiquitin ligase VCP (in yeast, cdc48) to MFN (in yeast, Fzo1) to regulate its stability (Ackema et al, 2014), and our results suggest that a similar mechanism may be relevant in humans (Fig. 3j). According to our data, ARF1 dysfunction impacts autophagy in general, not just mitophagy in particular, although this point requires future clarification (Fig. S3g and h). Importantly, mitophagy/autophagy seem not to be responsible for mitochondrial destabilization (Fig. S3i). As aberrant mitochondrial DNA release is prevented by reducing MFN levels (Fig. 3j), it is tempting to assume that type I IFN signaling would also be reduced.